# RECAST: Expanding the Boundaries of LLMs' Complex Instruction Following with Multi-Constraint Data

**Zhengkang Guo[1]\*** **Wenhao Liu[1,2]\*** **Mingchen Xie[2]** **Jingwen Xu[1]** **Zisu Huang[1]**
**Muzhao Tian[1]** **Jianhan Xu[2]** **Yuanzhe Shen[1]** **Qi Qian[1]** **Muling Wu[1]** **Xiaohua Wang[1]**

**Changze Lv[1]** **HeDa Wang[2]** **Hu Yao[2]** **Xiaoqing Zheng[1]†** **Xuanjing Huang[1]**

[1] College of Computer Science and Artificial Intelligence, Fudan University
[2] Xiaohongshu Inc.
 {zkguo24, whliu22}@m.fudan.edu.cn   zhengxq@fudan.edu.cn

## Abstract

Large language models (LLMs) are increasingly expected to tackle complex tasks, driven by their expanding applications and users' growing proficiency in crafting sophisticated prompts. However, as the number of explicitly stated requirements increases (particularly more than 10 constraints), LLMs often struggle to accurately follow such complex instructions, which limits their applicability in complex real-world scenarios. To the best of our knowledge, existing datasets do not exceed 10 constraints per instance. To address this challenge, we propose RECAST, an efficient and scalable framework for synthesizing datasets where each example incorporates far more constraints than those in existing benchmarks, aiming to challenge and extend the boundaries of models' ability to follow complex instructions. These constraints are extracted from real-world prompt-response pairs to ensure practical relevance. Using this framework, we construct RECAST-30K, a large-scale, high-quality dataset comprising 30k instances spanning 19 constraint types. Experimental results demonstrate that models fine-tuned on RECAST-30K substantially improve in following complex instructions while maintaining their general capabilities without degradation. Moreover, RECAST enables automatic verification of constraint satisfaction via rule-based validators for quantitative constraints and LLM-based validators for qualitative ones, the verifiability provided by RECAST enables the design of reward functions for reinforcement learning, which further boosts model performance on complex and challenging tasks. Code and data are available at `https://github.com/Thekey756/RECAST`.

## 1 Introduction

Large language models (LLMs) have demonstrated remarkable capabilities in solving various NLP tasks (Ahn et al., 2024; Chakrabarty et al., 2024; Gómez-Rodríguez & Williams, 2023; Huang & Chang, 2023; Jiang et al., 2024a; Zhao et al., 2023; Zheng et al., 2023) and are widely used in practical scenarios (Gu et al., 2024; Liu et al., 2024; Sun, 2023; Thirunavukarasu et al., 2023). However, they still struggle with complex multi-constraint scenarios that are prevalent in real-world applications, such as constitutional AI systems requiring adherence to multiple principles simultaneously (Bai et al., 2022; Findeis et al., 2025; Kundu et al., 2023; Kyrychenko et al., 2025), and enterprise assistants managing detailed business rules (Michele et al., 2025; Grohs et al., 2023; Robino, 2025).

Recently, LLM-based autonomous agents have attracted increasing attention from both academia and industry. Surveys such as Wang et al. (2024b) highlight their tremendous potential in social sciences, natural sciences, and engineering, which fundamentally relies on the ability of LLMs to follow complex instructions. As shown in Figure 1, when instruction complexity increases, with multiple explicit constraints in a single prompt, even advanced models like GPT-4o (Achiam et al., 2023)

---

\*These authors contributed equally.
†Corresponding author.

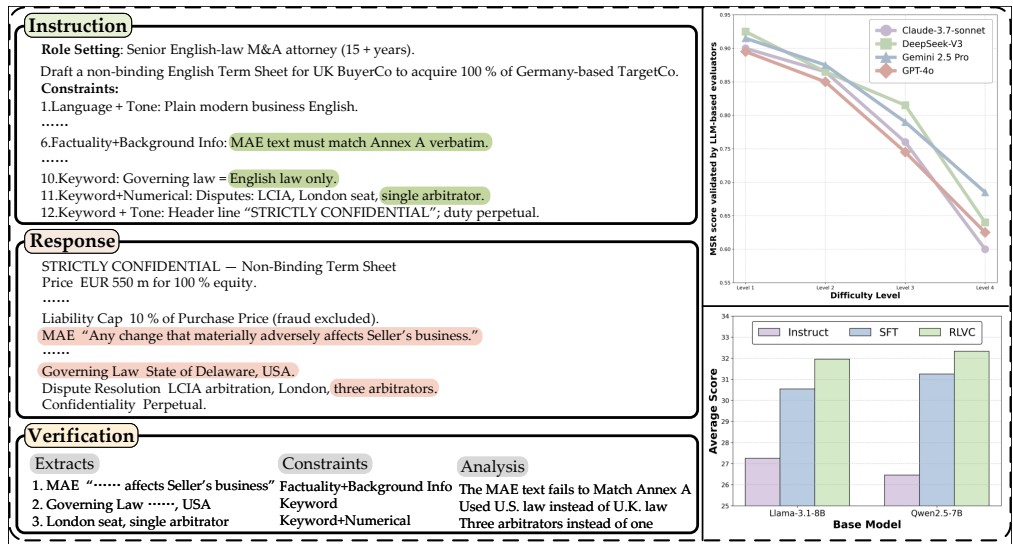

Figure 1: Challenges in complex instruction following and improved performance via RECAST-30k. Left: Real-world examples illustrating how LLMs fail to follow complex instructions. Top-right: Performance degradation of LLMs as the number of constraints increases. Bottom-right: Comparison of instruction-following performance between models fine-tuned on RECAST and their corresponding Instruct variants (e.g., Qwen2.5-7B-Instruct).

show marked performance degradation, which has been emphasized in multiple benchmarks (Ferraz et al., 2024; Jiang et al., 2024b; Qin et al., 2024; Wen et al., 2024). Therefore, advancing the complex instruction-following capabilities of LLMs is crucial for promoting their practical deployment.

Although prior studies have explored methods to evaluate and improve the instruction-following abilities of LLMs, existing research remains limited in scope. Works such as Jiang et al. (2024b) and Wen et al. (2024) evaluated LLMs' instruction-following performance and explored the use of LLMs themselves for evaluation. However, their datasets were manually constructed, lacked training sets, and thus fell short in terms of scalability for large-scale applications. While AutoIF (Dong et al., 2025) attempted to address data construction through an automated pipeline, it relied exclusively on code-verifiable constraints, which limits applicability to narrow domains and fails to capture broader, real-world use cases. More critically, most existing datasets impose only a small number of constraints (typically 3–5) and lack systematic evaluation benchmarks or training resources for complex instruction-following tasks. Previous automated approaches primarily rely on LLM-based instruction rewriting, which often yields homogenized constraints and limits scalability in generating large numbers of constraints. Consequently, they are inadequate for rigorously assessing and extending the capability boundaries of LLMs in realistic, complex application settings.

To address these limitations and advance research in this area, we propose RECAST, a scalable data-synthesis framework that constructs instruction-following datasets of unprecedented complexity. Unlike prior approaches, our pipeline systematically mines instruction-following signals from existing data, thereby improving data utilization and fully exploiting the training value of available resources. Through the integration of diverse, verifiable constraints, RECAST enables more rigorous evaluation and more effective enhancement of LLMs' ability to follow complex instructions.

Based on RECAST, we constructed a high-quality, diverse, and constraint-verifiable dataset, RECAST-30K, which encompasses a large number of constraints spanning diverse types. Using only 30K training samples, Llama3.1-8B-Base (Grattafiori et al., 2024) fine-tuned on RECAST-30K already outperforms other instruction-tuning datasets. Remarkably, it surpasses the corresponding instruct model—trained on a much larger-scale dataset—and even outperforms the substantially larger Llama-3.3-70B-Instruct in handling complex instruction-following scenarios.

Given that the constraints in RECAST-30K can be reliably verified using both rule-based and model-based validators, the dataset is inherently amenable to reinforcement learning. Building on this property, we design RLVC, which further enhances constraint compliance, yielding substantial

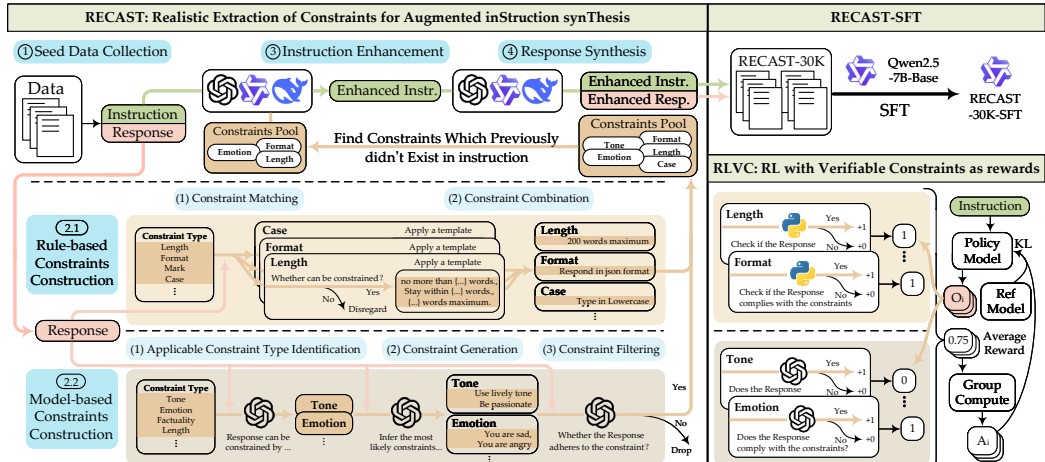

Figure 2: Overview of the RECAST framework and RLVC. Left: The RECAST pipeline generates complex instruction-following data through four steps: (1) seed data collection across diverse domains, (2) constraint construction with both rule-based and model-based verification methods, (3) instruction enhancement by integrating selected constraints, and (4) response synthesis ensuring constraints are satisfied. More detailed descriptions of the pipeline are provided in Appendix A. Top-right: Using RECAST-generated data to fine-tune LLMs through SFT. Bottom-right: RLVC framework leveraging constraint-specific verification to provide fine-grained rewards, guiding model optimization toward satisfying multiple constraints simultaneously.

improvements in multi-constraint satisfaction while preserving general capabilities. To summarize the highlights of our study:

- We propose **RECAST**, an efficient, scalable, and low-cost data-synthesis framework that constructs complex instruction-following datasets of unprecedented complexity, aiming to challenge and enhance state-of-the-art models' ability to follow complex instructions.

- We release **RECAST-30K**, a high-quality dataset purpose-built to benchmark and improve complex instruction-following performance.

- We further exploit the verifiable constraints in RECAST-30K to design **RLVC**, a reinforcement-learning approach that leverages constraint-specific reward signals for simultaneous multi-objective optimisation, thereby fully exploiting the supervision potential of the dataset.

## 2 REALISTIC EXTRACTION OF CONSTRAINTS FOR AUGMENTED INSTRUCTION SYNTHESIS

We present RECAST, a comprehensive framework for improving LLMs' ability to follow complex instructions. As shown in Figure 2, the RECAST pipeline operates through a three-stage process. First, we construct a rich constraint pool by extracting both rule-based and model-based constraints from seed data. Next, we synthesize enhanced instructions by selecting and naturally integrating appropriate constraints into original prompts. Finally, we generate consistent, constraint-compliant responses to these augmented instructions.

### 2.1 SEED DATA COLLECTION

We select Tülu 3 Persona IF (Lambert et al., 2024) as the seed dataset for constructing RECAST-30K. This dataset spans diverse scenarios and task types, including code generation, creative writing, factual question answering, etc., rendering it an appropriate foundation for constraint extraction. A more detailed analysis of the characteristics of this seed dataset is provided in Appendix B.1.

## 2.2 Constraint Pool Construction

To effectively model the multifaceted nature of real-world instructions, we develop a comprehensive constraint pool that captures both objective and subjective requirements. We categorize constraints into two complementary types based on their verification mechanisms, enabling precise assessment of model outputs against diverse criteria.

**Rule-based Constraints Construction.** Rule-based constraints represent objective requirements that can be verified through deterministic methods, including structural elements (paragraph count), lexical specifications (keyword inclusion/exclusion), and quantitative parameters (word limits). These constraints are particularly valuable as they provide unambiguous verification signals for model training and evaluation. To systematically extract these constraints, we implement nine specialized rule-based extractors (detailed in Appendix E.1) that analyze responses to identify verifiable properties. These extractors recognize specific syntactic patterns, keyword frequencies, numerical parameters, and structural characteristics that can be programmatically verified. This programmatic approach ensures that each rule-based constraint has a corresponding verification method, providing clear training signals for models to learn specific response characteristics.

**Model-based Constraints Construction.** Model-based constraints encompass subjective requirements that necessitate semantic understanding or qualitative judgment, such as stylistic elements (formality level), tonal qualities (politeness), and content characteristics (persuasiveness). These constraints are crucial for capturing the nuanced aspects of human communication that cannot be reduced to simple rules. Our construction process follows three key steps: First, we analyze each response from the seed datase to determine which constraint types from our taxonomy of 10 categories (detailed in Appendix E.2) are applicable, ensuring relevance to the specific content. Second, we employ LLMs to generate multiple concrete constraint instances for each applicable type, creating a comprehensive initial pool of model-based constraints. Third, we implement a filtering process to verify that each constraint is actually satisfied by its corresponding response, removing those that cannot be fulfilled. To validate the accuracy of filtering process, we conducted human evaluation on randomly selected constraint-response pairs, confirming that our methodology successfully identifies and retains only appropriate, verifiable constraints (detailed results in Appendix C.1).

Through this dual-mode construction approach, we create a comprehensive constraint pool that captures both the objective, rule-verifiable aspects and the subjective, semantically-rich dimensions of instruction following.

## 2.3 Constraint-Augmented Instruction Synthesis

After establishing a verified constraint pool, we develop a three-stage process to create enhanced instructions that effectively incorporate appropriate constraints. This process aims to ensure that the resulting instructions maintain natural language quality while systematically integrating multiple constraints.

**Constraint Selection.** We employ LLMs to select a coherent subset of constraints from the pool that are relevant and applicable to each original instruction. This selection process ensures that only contextually appropriate constraints are considered for integration, avoiding the inclusion of irrelevant or contradictory requirements.

**Instruction Enhancement.** Once appropriate constraints are identified, we task multiple LLMs with integrating the selected constraints into the original instruction. Each model produces an enhanced instruction where constraints are incorporated naturally and coherently into the text.

**Optimal Instruction Selection.** To ensure quality and consistency, we implement a majority voting mechanism wherein multiple LLMs evaluate and rank the candidate integrated instructions. The evaluation criteria include linguistic fluency, semantic coherence, and constraint completeness. We select the highest-ranked instruction as the final constraint-augmented instruction. We also performed random sampling on optimal instruction selection for human evaluation, the results of human evaluation are shown in Appendix C.2.

## 2.4 INSTRUCTION-CONSISTENT RESPONSE SYNTHESIS

To address potential inconsistencies between constraint-augmented instructions and original responses in the seed dataset, we implement a two-phase process to ensure alignment while maintaining high response quality.

**Diverse Response Generation.** We generate multiple candidate responses to each constraint-augmented instruction using distinct LLMs. This multi-model approach introduces diversity into the candidate pool and mitigates potential biases inherent to any single model. Each model generates responses independently, ensuring a broad representation of possible approaches to satisfying the complex instruction.

**Response Quality Assessment.** We adopt a majority voting scheme considering constraint adherence, accuracy, conciseness, etc., to select the best response for each instruction. The chosen response is added to the dataset with its enhanced instruction. Human evaluation on sampled pairs (Appendix C.3) confirms the high quality of the selected responses.

This methodical approach to generation and selection ensures that our final dataset comprises constraint-rich and high-quality instruction–response pairs that demonstrate effective navigation of complex, multi-constraint scenarios. Each pair is annotated with its full set of constraints and corresponding validation method, facilitating downstream application and evaluation.

## 3 REINFORCEMENT LEARNING WITH VERIFIABLE CONSTRAINTS AS REWARDS

Since the constraints in RECAST are equipped with corresponding verification methods, the dataset is inherently amenable to reinforcement learning. We therefore introduce RLVC, which leverages the verifiable nature of constraints to provide fine-grained reward signals during policy optimization. Building on SFT-trained models, we further apply RL to maximize the training value of RECAST-30K for enhancing complex instruction-following capabilities. This approach enables more targeted feedback on constraint satisfaction, thereby improving models' ability to simultaneously address multiple complex requirements.

### 3.1 CONSTRAINT VERIFICATION MECHANISMS

We implement a dual-mode verification scheme that enables precise evaluation of constraint satisfaction. Each constraint in an instruction is assessed through either rule-based or LLM-based verification methods. For rule-based constraints, we employ rule-based validators $V_{\text{rule-based}}(x, y, c_i)$ that programmatically verify objective requirements through deterministic procedures. For model-based constraints that require evaluation of subjective qualities, we leverage LLM-based validators $V_{\text{model-based}}(x, y, c_i)$. These validators assess more nuanced requirements that resist codification into explicit codes. We formalize the constraint verification function for an instruction $x$ with response $y$ and constraint $c_i$ as:

$$f(x, y, c_i) = \begin{cases} V_{\text{rule-based}}(x, y, c_i) & \text{if } c_i \text{ is rule-based} \\ V_{\text{model-based}}(x, y, c_i) & \text{if } c_i \text{ is model-based} \end{cases} \tag{1}$$

where both verification functions return binary values indicating constraint satisfaction (1) or violation (0).

### 3.2 VERIFIABLE CONSTRAINTS AS REWARD SIGNALS

The key innovation of RLVC lies in its exploitation of RECAST's constraint verifiability feature. Traditional reinforcement learning approaches typically rely on a single, holistic reward signal that fails to identify which specific constraints are violated. In contrast, RECAST-30K provides individually verifiable constraints, enabling us to design a more informative reward mechanism. For an instruction $x$ containing multiple constraints $C = \{c_1, c_2, ..., c_n\}$, we define the reward for a generated response $y$ as the average satisfaction rate across all constraints:

$$R(x, y) = \frac{1}{|C|} \sum_{i=1}^{|C|} f(x, y, c_i) \tag{2}$$

This design provides an independent reward channel for each constraint, offering fine-grained feedback that guides the model toward simultaneously satisfying multiple constraints. Our RLVC framework combines verifiable constraints as reward signals with Group Relative Policy Optimization (GRPO) (Shao et al., 2024). Specifically, RLVC leverages the fine-grained feedback enabled by constraint verifiability and optimizes policies via GRPO's comparative learning approach. This integration provides the model with clear guidance on improving constraint satisfaction across diverse instruction types. For completeness, we present the detailed formulation of GRPO, including its optimization objective, in Appendix D.5.

## 4 EXPERIMENTS

### 4.1 SETUPS

**Evaluation Benchmarks.** We constructed RECAST-Test, a hierarchical benchmark with four difficulty levels defined by increasing constraint complexity, enabling fine-grained evaluation across tasks of varying difficulty. Crucially, RECAST-Test contains substantially more constraints than prior benchmarks, providing a more rigorous assessment of LLMs' complex instruction-following capabilities. Details of its construction and statistics are in Appendix B.4. To further assess generalization, we also evaluate RECAST on four external benchmarks (Appendix D.2).

**Evaluation Metrics.** For evaluation metrics, we use the Hard Constraint Satisfaction Rate (HSR) as the primary metric, quantifying the model's ability to satisfy all specified constraints in an instruction simultaneously. The HSR is calculated as follows:

$$\text{HSR} = \frac{1}{|D|} \sum_{(x,y) \in D} \prod_{c_i \in C(x)} f(x, y, c_i) \tag{3}$$

where $D$ represents the evaluation dataset, $C(x)$ is the set of constraints for instruction $x$, and $f(x, y, c_i)$ is the verification function for constraint $c_i$.

Additionally, we define the following sub-metrics for different constraint types: (1) *Rule-based Constraint Satisfaction Rate (RSR)*, which measures the HSR score for constraints evaluated using rule-based methods; (2) *Model-based Constraint Satisfaction Rate (MSR)*, which measures the HSR score for constraints evaluated using LLMs; and (3) *Overall Constraint Satisfaction Rate (OSR)*, which quantifies the HSR of all constraints, both rule-based and model-based, that are successfully satisfied simultaneously. For the FollowBench evaluation, we similarly adopt the Hard Constraint Satisfaction Rate defined in the original benchmark. For all LLM-based evaluations, unless otherwise specified, we default to using GPT-4o (Achiam et al., 2023) for assessment.

**Baselines.** For our baseline comparisons, we primarily select eight high-quality, open-source complex instruction fine-tuning datasets. A detailed description of the baselines can be found in Appendix D.1. Additionally, we evaluated the performance of some of the leading large models on the benchmark we constructed.

**Settings.** For all experiments, we choose Qwen2.5-7B (Yang et al., 2024) and Llama-3.1-8B (Grattafiori et al., 2024) as the base models. The same experimental setup is used for SFT across all datasets. We use the model after SFT as the base model of RLVC for training. Specific experimental details can be found in Appendix D.

### 4.2 MAIN RESULTS

#### 4.2.1 EVALUATION ON RECAST-TEST

Table 1 presents comprehensive results from our multi-level constraint satisfaction benchmark. Analysis of these findings reveals three results:

**Performance Degradation with Increasing Constraint Complexity.** Across all models, performance consistently declines from Level 1 to Level 4, underscoring the difficulty of satisfying multiple constraints simultaneously. Rule-based constraints prove more challenging than model-based ones. Gemini-2.5-Pro shows the best performance with a **39.75**% average satisfaction rate. Overall, complex instruction following remains difficult for current LLMs, highlighting the need for further advancement.

**Effectiveness of RECAST-30K for Instruction Following.** Models fine-tuned on RECAST-30K demonstrate substantial improvements in constraint satisfaction capabilities. The RECAST-30K-SFT variant achieves a **31.25**% average satisfaction rate on Qwen-2.5-7B, significantly outperforming both instruction-tuned models

Table 1: RECAST-Test results across four difficulty levels and overall average. RSR, MSR, and OSR denote the rule-based, model-based, and overall components of the holistic success rate (HSR), respectively. The best result in each column is highlighted in bold, while the second-highest result is indicated with underline formatting. All reported values are percentages, with the percent sign omitted for brevity.

| Models | Level 1 | | | Level 2 | | | Level 3 | | | Level 4 | | | Average |
|---|---|---|---|---|---|---|---|---|---|---|---|---|---|
| | MSR | RSR | OSR | MSR | RSR | OSR | MSR | RSR | OSR | MSR | RSR | OSR | |
| Claude-3.7-sonnet | 90.00 | 15.50 | 13.50 | 86.50 | 10.00 | 7.00 | 76.00 | 5.50 | 5.00 | 60.00 | 6.50 | 4.50 | 31.67 |
| DeepSeek-V3 | 92.50 | 24.00 | 21.50 | 86.50 | 14.50 | 11.50 | 81.50 | 9.00 | 7.50 | 64.00 | 8.50 | 5.00 | 35.50 |
| Gemini-2.5-Pro | 91.50 | 25.50 | 22.00 | 87.50 | 21.00 | 19.00 | 79.00 | 16.50 | 13.00 | 68.50 | 20.00 | 13.50 | 39.75 |
| GPT-4o | 89.50 | 24.50 | 22.50 | 85.00 | 15.00 | 10.50 | 74.50 | 7.50 | 6.00 | 62.50 | 9.00 | 7.00 | 34.46 |
| Qwen3-235B-A22B | 96.00 | 25.50 | 25.00 | 87.00 | 13.50 | 10.00 | 83.00 | 12.00 | 10.00 | 71.00 | 7.00 | 5.50 | 37.13 |
| Qwen2.5-72B-Instruct | 88.00 | 23.50 | 20.50 | 79.50 | 16.50 | 12.00 | 70.00 | 8.50 | 6.50 | 64.50 | 6.50 | 4.50 | 33.38 |
| Qwen2.5-7B-Instruct | 83.50 | 21.50 | 18.50 | 69.00 | 12.50 | 7.50 | 51.00 | 3.50 | 2.00 | 40.50 | 5.00 | 3.00 | 26.46 |
| Llama-3.3-70B-Instruct | 79.00 | 25.00 | 20.00 | 56.00 | 14.50 | 9.00 | 43.00 | 9.00 | 3.00 | 30.00 | 6.00 | 2.50 | 24.75 |
| Llama-3.1-8B-Instruct | 86.50 | 18.50 | 15.50 | 73.00 | 12.50 | 8.00 | 53.50 | 8.50 | 4.50 | 37.00 | 6.00 | 3.50 | 27.25 |
| **Llama-3.1-8B** | | | | | | | | | | | | | |
| Conifer | 73.00 | 15.00 | 11.50 | 54.00 | 10.00 | 3.00 | 45.00 | 4.50 | 2.50 | 32.50 | 0.50 | 0.50 | 21.00 |
| Crab | 59.00 | 13.00 | 6.00 | 33.50 | 8.00 | 3.00 | 24.50 | 3.00 | 0.50 | 12.00 | 2.50 | 0.50 | 13.79 |
| I-SHEEP | 66.00 | 14.00 | 8.75 | 43.75 | 9.00 | 3.00 | 34.75 | 3.75 | 1.50 | 22.25 | 1.50 | 0.50 | 17.40 |
| MUFFIN | 34.50 | 6.00 | 1.50 | 19.00 | 4.50 | 2.00 | 10.00 | 1.00 | 0.00 | 13.50 | 1.50 | 0.00 | 7.79 |
| ShareGPT | 71.00 | 16.00 | 10.50 | 49.00 | 5.00 | 2.00 | 38.00 | 3.00 | 0.50 | 27.50 | 2.50 | 1.00 | 18.83 |
| Evol-Instruct | 62.50 | 17.00 | 10.50 | 42.50 | 6.50 | 3.00 | 28.00 | 3.50 | 0.50 | 20.50 | 2.50 | 1.00 | 16.50 |
| Suri | 3.00 | 3.00 | 0.00 | 1.00 | 2.50 | 0.00 | 0.50 | 0.00 | 0.00 | 0.50 | 0.00 | 0.00 | 0.88 |
| Tülu 3 Persona IF | 83.00 | 22.00 | 19.50 | 70.50 | 9.00 | 5.50 | 55.50 | 8.00 | 2.50 | 42.00 | 4.00 | 2.00 | 26.96 |
| RECAST-30K-SFT | 86.00 | 21.00 | 18.50 | 78.50 | 15.00 | 9.50 | 63.50 | 6.00 | 4.00 | 52.00 | 8.00 | 4.50 | 30.54 |
| RECAST-30K-RLVC | 83.50 | 21.00 | 17.00 | 79.50 | 17.50 | 12.50 | 64.00 | 11.50 | 7.50 | 53.00 | 10.00 | 6.50 | 31.96 |
| **Qwen-2.5-7B** | | | | | | | | | | | | | |
| Conifer | 71.00 | 16.00 | 14.00 | 58.50 | 9.00 | 5.50 | 45.50 | 6.00 | 3.50 | 33.50 | 3.00 | 1.00 | 22.21 |
| Crab | 57.50 | 14.00 | 7.50 | 30.00 | 7.00 | 3.00 | 20.50 | 5.00 | 1.00 | 15.50 | 5.50 | 1.50 | 14.00 |
| I-SHEEP | 56.00 | 12.50 | 7.50 | 35.50 | 6.00 | 1.00 | 26.00 | 3.50 | 0.00 | 18.00 | 1.50 | 0.00 | 13.96 |
| MUFFIN | 61.00 | 17.00 | 10.00 | 46.00 | 7.00 | 4.50 | 25.50 | 3.50 | 1.00 | 25.00 | 3.50 | 1.00 | 17.13 |
| ShareGPT | 70.50 | 13.50 | 11.00 | 46.50 | 9.50 | 4.00 | 34.50 | 3.50 | 2.50 | 28.00 | 4.50 | 1.50 | 19.13 |
| Evol-Instruct | 63.50 | 14.50 | 8.00 | 41.00 | 8.00 | 3.00 | 27.00 | 2.50 | 1.00 | 22.50 | 4.50 | 2.00 | 16.46 |
| Suri | 0.50 | 4.50 | 0.00 | 0.50 | 2.00 | 0.00 | 0.00 | 0.50 | 0.00 | 0.00 | 0.00 | 0.00 | 0.67 |
| Tülu 3 Persona IF | 88.00 | 21.00 | 19.50 | 69.00 | 13.00 | 7.00 | 48.50 | 4.50 | 1.50 | 45.00 | 4.00 | 1.00 | 26.83 |
| RECAST-30K-SFT | 87.50 | 18.50 | 14.50 | 76.50 | 17.00 | 13.00 | 66.50 | 11.50 | 6.50 | 54.00 | 5.00 | 4.50 | 31.25 |
| RECAST-30K-RLVC | 85.50 | 22.50 | 16.50 | 78.00 | 19.00 | 12.50 | 65.50 | 10.50 | 4.50 | 55.50 | 11.00 | 7.00 | 32.33 |

and alternative fine-tuning approaches. This enhancement underscores the effectiveness of our constraint-focused dataset in developing robust instruction-following capabilities, particularly for navigating complex, multi-constraint scenarios.

**Performance Enhancement Through RLVC Optimization.** The application of RLVC further enhances model performance across all difficulty levels. RECAST-30K-RLVC variants consistently outperform their SFT counterparts, achieving a **32.33**% average satisfaction rate on Qwen-2.5-7B. This improvement is particularly pronounced in higher difficulty levels (Levels 3-4), where constraint interactions become more complex. These gains confirm that our constraint-specific reinforcement learning approach effectively optimizes for simultaneous satisfaction of multiple requirements, addressing a key challenge in practical complex instruction following.

These results clearly demonstrate that specialized fine-tuning approaches—particularly our RECAST-30K dataset and RLVC optimization framework—significantly enhance LLMs' capability to satisfy multiple complex constraints simultaneously. This improvement addresses a critical gap in current instruction-following systems and enables more reliable performance in constraint-rich applications.

**Comparison with Divide-Verify-Refine (DVR).** We further compare our models with the inference-time instruction-following enhancement method Divide-Verify-Refine (DVR) (Zhang et al., 2025) on RECAST-Test. Since DVR in its original form only leverages rule-based verifiers, we report the Rule-based Constraint Satisfaction Rate (RSR) for a fair comparison. As shown in Table 2, applying DVR at inference time on top of Llama-3.1-8B-Instruct and Qwen2.5-7B-Instruct leads to substantially lower RSR than our RECAST-30K-SFT models, which consistently achieve higher RSR across all difficulty levels.

These results show that, for both Llama-3.1-8B and Qwen2.5-7B, RECAST-30K-SFT achieves substantially higher average RSR than their DVR-enhanced counterparts: **12.5**% vs. 7.0% on Llama (**+5.5**% points) and **13.0**% vs. 6.5% on Qwen (**+6.5**% points). The gains are consistent across all difficulty levels (Levels 1–4), typically ranging from 3% to 8% absolute points. This suggests that our training-time RECAST-30K pipeline provides complementary benefits beyond inference-time self-alignment, leading to more robust multi-constraint satisfaction.

### 4.2.2 EVALUATION ON ADDITIONAL INSTRUCTION-FOLLOWING BENCHMARKS.

Table 3 reports results on two widely used instruction-following benchmarks, IFEVAL and FollowBench. RECAST-30K-SFT already surpasses all baseline methods, and RECAST-30K-RLVC further improves per-

Table 2: RSR results of Divide-Verify-Refine (DVR) compared with our RECAST-30K-SFT models on RECAST-Test. We report Rule-based Constraint Satisfaction Rate (RSR) across four difficulty levels and their average. The best result in each column is highlighted in bold, while the second-highest result is indicated with underline formatting. All reported values are percentages, with the percent sign omitted for brevity.

| Model | Level 1 | Level 2 | Level 3 | Level 4 | Avg. |
|---|---|---|---|---|---|
| Llama-3.1-8B-Instruct + DVR | 14.0 | 9.5 | 2.0 | 2.5 | 7.0 |
| Qwen2.5-7B-Instruct + DVR | 11.0 | 8.5 | 4.5 | 2.0 | 6.5 |
| RECAST-30K-SFT (Ours, Llama base) | **21.0** | 15.0 | 6.0 | **8.0** | 12.5 |
| RECAST-30K-SFT (Ours, Qwen base) | 18.5 | **17.0** | **11.5** | 5.0 | **13.0** |

Table 3: Results on instruction-following benchmarks (IFEVAL and FollowBench). IFEVAL uses Prompt-Level Loose Accuracy (*Pr. (L)*), while FollowBench uses Hard Satisfaction Rate (*HSR*). The best result in each column is highlighted in bold, while the second-highest result is indicated with underline formatting. All reported values are percentages, with the percent sign omitted for brevity.

| Models | Llama3.1-8B | | | QWEN2.5-7B | | |
|---|---|---|---|---|---|---|
| | IFEVAL | Followbench | Avg. | IFEVAL | Followbench | Avg. |
| | *Pr. (L)* | *HSR* | | *Pr. (L)* | *HSR* | |
| **Conifer** | 43.62 | 44.69 | 44.16 | 44.73 | 52.27 | 48.50 |
| **Crab** | 42.14 | 41.90 | 42.02 | 44.55 | 44.50 | 44.53 |
| **I-SHEEP** | 32.90 | 29.48 | 31.19 | 36.60 | 35.38 | 35.99 |
| **MUFFIN** | 24.95 | 16.78 | 20.87 | 36.78 | 42.15 | 39.47 |
| **ShareGPT** | 46.40 | 45.41 | 45.91 | 47.13 | 49.06 | 48.10 |
| **Evol-Instruct** | 42.70 | 36.60 | 39.65 | 45.29 | 46.34 | 45.82 |
| **Suri** | 18.30 | 4.02 | 11.16 | 12.75 | 5.59 | 9.17 |
| **Tülu 3 Persona IF** | 73.94 | 55.44 | 64.69 | 73.94 | 56.94 | 65.44 |
| **RECAST-30K-SFT** | 76.34 | 57.10 | 66.72 | 73.38 | 59.60 | 66.49 |
| **RECAST-30K-RLVC** | **77.39** | **61.76** | **69.58** | **74.01** | **63.23** | **68.62** |

formance, achieving the best results on both Llama3.1-8B and QWEN2.5-7B. These consistent gains on out-of-domain benchmarks demonstrate that RECAST not only improves complex constraint-following but also generalizes effectively to traditional instruction-following tasks, rather than merely overfitting to the constraint patterns seen during training.

### 4.2.3 GENERAL CAPABILITY EVALUATION

In addition, we evaluate RECAST on two general capability benchmarks (Table 4), where RECAST-30K-RLVC achieves average scores of 34.64% on Llama3.1-8B and 38.39% on QWEN2.5-7B. This indicates that while RECAST substantially improves performance on complex instruction following, it also preserves competitive performance on reasoning-oriented benchmarks like MUSR and knowledge-focused evaluations such as GPQA.

## 5 RELATED WORK

**Complex Instruction Datasets Construction.** Recent work has proposed various methods for constructing complex instruction datasets. Xu et al. (2024) proposed Evol-Instruct, which increases difficulty through iterative rewriting. Sun et al. (2024) introduced Conifer, which generates multi-level constraints with GPT-4. An et al. (2025) proposed ULTRAIF, which decomposes and recombines instructions, and Liu et al. (2025) introduced AIR, which adds constraints via a model-judge cycle. Other approaches include MUFFIN by Lou et al. (2023), which expands input features, and CRaB by Qi et al. (2024), which infers constraints via back-translation. In contrast, RECAST mines diverse constraint types directly from seed responses and integrates explicit verification for both rule-based and model-based constraints, addressing limitations in constraint complexity, diversity, and quality assurance.

**Complex Instruction Following Capabilities Improvement.** Recent algorithmic innovations have significantly advanced instruction following capabilities. Dong et al. (2025) proposed AutoIF, which implements a self-dialogue generation process with execution-based verification, filtering training data through executable feedback.

Table 4: Results on general capability benchmarks (GPQA and MUSR). All metrics use standard Accuracy (*Acc*), measuring the percentage of correct responses. The best result in each column is highlighted in bold, while the second-highest result is indicated with underline formatting. All reported values are percentages, with the percent sign omitted for brevity.

| Models | Llama3.1-8B | | | QWEN2.5-7B | | |
| | GPQA | MUSR | Avg. | GPQA | MUSR | Avg. |
| | *Acc* | *Acc* | | *Acc* | *Acc* | |
|---|---|---|---|---|---|---|
| **Conifer** | 27.61 | 37.64 | 32.63 | 31.22 | 45.54 | 38.38 |
| **Crab** | **32.48** | 40.26 | 36.37 | 31.89 | **48.74** | **40.32** |
| **I-SHEEP** | 28.33 | 36.30 | 32.32 | 29.95 | 43.18 | 36.57 |
| **MUFFIN** | 26.64 | 33.69 | 30.17 | 29.07 | 37.93 | 33.50 |
| **ShareGPT** | 30.02 | 40.16 | **35.09** | 31.20 | 44.95 | 38.08 |
| **Evol-Instruct** | 28.41 | **41.75** | 35.08 | **32.55** | 47.41 | 39.98 |
| **Suri** | 26.09 | 33.40 | 29.75 | 26.60 | 41.79 | 34.20 |
| **Tülu 3 Persona IF** | 28.64 | 38.68 | 33.66 | 31.06 | 44.13 | 37.60 |
| **RECAST-30K-SFT** | 26.77 | 37.39 | 32.08 | 30.02 | 39.91 | 34.97 |
| **RECAST-30K-RLVC** | 30.81 | 38.47 | 34.64 | 29.84 | 46.94 | 38.39 |

Wang et al. (2024a) introduced RNR, which extracts roles and rules from existing instructions to generate rule-compliant responses, while Cheng et al. (2025) proposed SPaR, which employs self-play tree search refinement through multiple rounds of model self-adversarial interaction to optimize response quality. The approach in (He et al., 2024) utilizes discriminative generation, applying a discriminator model to filter generated samples for higher-quality supervision. Unlike prior methods, we leverage RECAST-30K's quantitative and qualitative constraints, using multi-channel rewards in RLVC to treat each constraint as a separate optimization target, significantly improving accuracy and consistency in complex instruction following.

**Reinforcement Learning with Verifiable Rewards.** A closely related line of work studies reinforcement learning with verifiable rewards (RLVR) for LLMs (Lambert et al., 2024; Su et al., 2025; Wang et al., 2025; Pyatkin et al., 2025). These methods replace or complement scalar-valued reward models with rewards obtained from rule-based or model-based verifiers, and have demonstrated strong gains in domains where answer correctness can be automatically checked. Lambert et al. (2024) and Wang et al. (2025) apply verifiable rewards to math, code, and other tasks with deterministic checkers, while Su et al. (2025) extends RLVR to broader domains and shows that verifiable rewards can serve as a bridge between different task families. Pyatkin et al. (2025) further investigates how verifiable reward mechanisms generalize to multi-constraint and format-constrained instruction following. Most of these works, however, still assume a single reward channel, either purely rule-based or purely model-based. By contrast, RLVC formulates rewards at the per-constraint level and explicitly combines rule-based and model-based verifiers within a unified framework, yielding richer partial-credit signals for complex instruction-following tasks.

## 6 CONCLUSION

We present RECAST, a scalable, low-cost framework for constructing complex instruction-following datasets through systematic mining and verification of rule- and model-based constraints. Using this, we release RECAST-30K, enabling smaller models (e.g., Llama3.1-8B) to outperform larger instruct-tuned models. Leveraging constraint verifiability, we propose RLVC, a reinforcement learning method using fine-grained, constraint-specific rewards to enhance performance. Experiments show RECAST-trained models excel on our hierarchical benchmark, generalize to IFEVAL and FollowBench, and remain competitive on general tasks (GPQA, MUSR). RECAST advances automated data construction by offering scalable, verifiable supervision, improving constraint diversity, complexity, and quality, and providing practical resources for more controllable LLMs, and advancing their real-world applications.

## ETHICS STATEMENT

This work focuses on constructing complex instruction following dataset and training methods for large language models (LLMs). All data used in RECAST-30K is derived from publicly available, non-sensitive sources and does not involve private or personally identifiable information. To mitigate risks of bias and harmful outputs, we systematically design rule-based and model-based constraints with explicit verification mechanisms, ensuring high-quality and safe training signals. While our framework substantially improves models' ability to follow complex, multi-constraint instructions, we recognize that enhanced controllability can also be misused (e.g., for

generating manipulative or deceptive content). We therefore emphasize that RECAST and RLVC are intended for advancing the reliability and transparency of LLMs in research and socially beneficial applications. We encourage responsible use and further auditing to address potential fairness, bias, and misuse concerns.

## REPRODUCIBILITY STATEMENT

We provide detailed descriptions to facilitate reproducibility of our work. In Section 2, we introduce the design and methodology of RECAST, with further details of the data construction pipeline presented in Appendix A. Additional implementation specifications, including API usage for data generation, are described in Appendix D.3. The supervised fine-tuning (SFT) setup, including hyperparameters and engineering resources, is documented in Appendix D.4. For reinforcement learning with verifiable constraints (RLVC), we describe the reward function design in Section 3, and provide theoretical formulations, implementation resources, and training dynamics in Appendix D.5. If accepted, we will release the code, data, and trained models on GitHub to further support reproducibility.

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

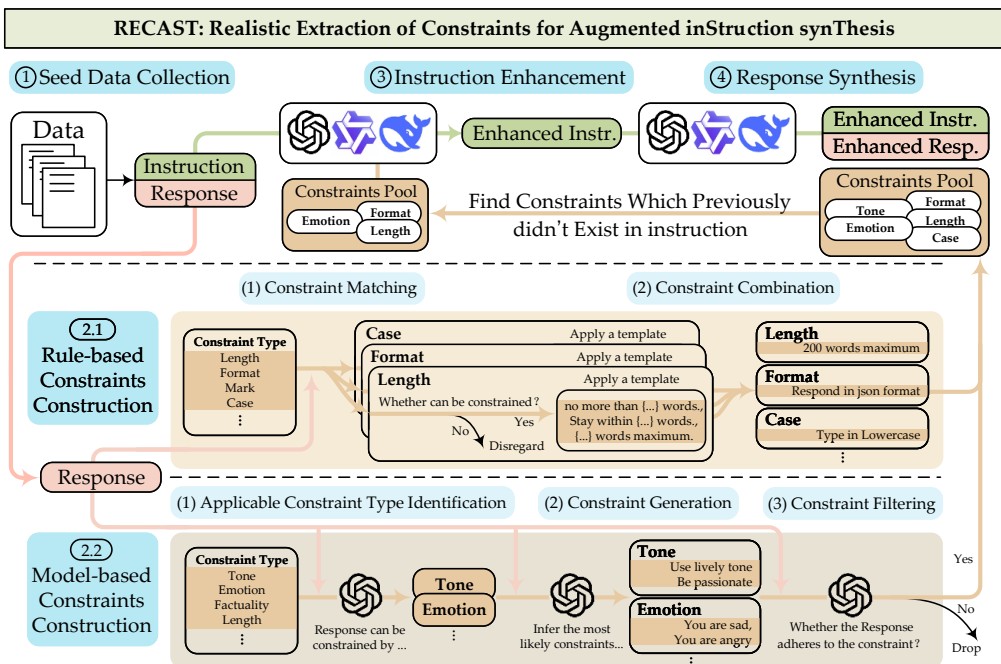

Figure 3: Overview of the RECAST framework. The RECAST pipeline generates complex instruction-following data through four steps: (1) seed data collection across diverse domains, (2) constraint construction with both rule-based and model-based verification methods, (3) instruction enhancement by integrating selected constraints, and (4) response synthesis ensuring constraints are satisfied.

## A DETAILED DESCRIPTION OF THE RECAST DATA CONSTRUCTION PIPELINE

To facilitate future research in complex instruction following and to support reproducibility, we provide a comprehensive description of the RECAST data construction pipeline in this appendix. The following exposition is presented in conjunction with Figure 3, which illustrates the pipeline workflow and should help readers better understand the individual steps. We believe this detailed methodology will serve as a valuable reference for future work on dataset construction and evaluation.

The pipeline begins with the Tülu 3 Persona IF dataset (Lambert et al., 2024) as the seed dataset, which contains instruction–response pairs. Based on these responses, we construct both rule-based and model-based constraints, which can be processed in parallel.

For rule-based constraints, we define multiple types (see Table 11), each associated with a validation function (rule-based validator). These functions determine whether a response satisfies the conditions for generating a rule-based constraint. When a condition is met, a matching template from our constraint template set (Appendix E.3) is selected to generate the corresponding constraint. For example, if a response is validated as JSON, we may select the template "Format your answer as valid '"' and instantiate it with "json" to form "Format your answer as valid 'json'" (Step 2.1 in Figure 2).

For model-based constraints, we define multiple types (see Table 12). An LLM first analyzes each response to identify applicable constraint types (Applicable Constraint Type Identification, Step 2.2 in Figure 2). Another prompt is then issued to generate specific constraints for each response (Step 2.2 in Figure 2; see prompt in Figure 22). Since some generated constraints may be spurious, we apply constraint filtering to retain only those actually satisfied by the response (Step 2.2 in Figure 2).

After constructing both rule-based and model-based constraints, we form a constraint pool for each response. An LLM then compares the original instruction with the constraint pool to identify missing constraints. These are incorporated into the instruction through rewriting, producing an Enhanced Instruction. To ensure rewrite quality, multiple LLMs are queried to generate candidate rewrites, followed by majority voting (Table 8) to select the best version.

Finally, given the enhanced instructions, we use multiple LLMs to generate responses. This procedure increases response diversity while ensuring alignment with the enhanced instructions. Majority voting is again applied to

select the best candidate, yielding an Enhanced Response. The resulting RECAST-30K dataset thus consists of Enhanced Instruction–Enhanced Response pairs, systematically constructed through this pipeline.

# B    DATASET ANALYSIS

In this section, we first introduce the characteristics of the seed dataset Tülu 3 Persona IF (Lambert et al., 2024) to justify its selection as the basis for our work, and then provide a detailed analysis of RECAST, the dataset we constructed for improving complex instruction following, with a particular focus on the characteristics of the constraints we designed and their distribution across different types.

## B.1    SEED DATASET ANALYSIS

The seed dataset for our RECAST-30K is Tülu 3 Persona IF, a subset of Tülu 3 containing 30K prompt–response pairs. We chose this dataset as our starting point because it spans a wide range of domains and includes responses rich in content—making it especially well-suited for constructing diverse and realistic constraints in our RECAST pipeline.

To further illustrate the breadth of task coverage in our seed dataset, we provide below a breakdown of the domain distribution. This diversity is a key factor that enables our framework to generate a wide variety of instruction-following tasks with meaningful, verifiable constraints. This domain-level diversity ensures that our RECAST-30K dataset supports constraint generation across a broad spectrum of real-world scenarios.

Our pipeline method is not limited to the currently used seed dataset; it is equally applicable to other corpora containing rich information. We chose this dataset primarily for ease of engineering implementation.

## B.2    RECAST-30K

To rigorously evaluate the ability of language models to follow complex, constraint-rich instructions, we construct a dataset consisting of **29,939** examples, which we refer to as **RECAST-30K**. RECAST-30K stands out for its high density of constraints, setting it apart from existing instruction datasets.

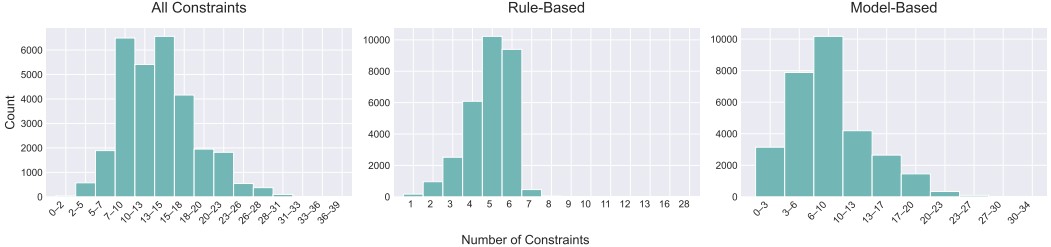

Figure 4: Constraint Count Distribution of RECAST-30K.

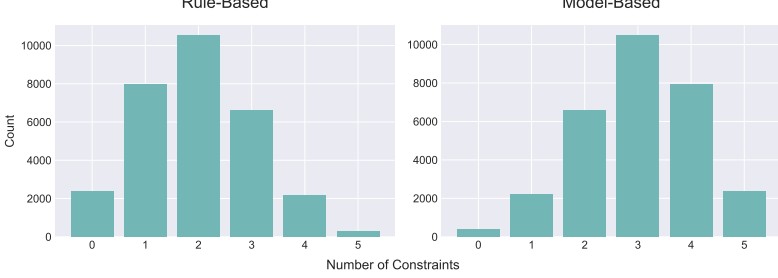

Figure 5: Constraint Count Distribution of Train Set with 5 Constraints.

### B.2.1    CONSTRAINT DENSITY

Each instruction in RECAST-30K is enriched with an average of **13.4** constraints drawn from different types. Among these, an average of **8.6** constraints are generated via large language models (model-based constraints),

Table 5: Domain distribution of the Tülu 3 Persona IF seed dataset.

| Domain | Number of Instances |
|---|---|
| Storytelling | 2334 |
| Social Media Post | 740 |
| Advertising Copy | 1196 |
| Scriptwriting & Dialogue | 616 |
| Poetry | 278 |
| Essay/Blog/Review | 3390 |
| Lyrics | 93 |
| Finance | 643 |
| Legal | 612 |
| Programming & Technology | 1376 |
| Medical & Health | 917 |
| Education | 4220 |
| Daily Life | 981 |
| Internet/IT | 126 |
| Data Analysis | 997 |
| Translation | 147 |
| Summarization | 992 |
| Information Retrieval | 912 |
| Science Popularization | 1556 |
| News Reporting | 610 |
| Entertainment | 294 |
| Business & Management | 1296 |
| Politics & International Affairs | 672 |
| Scientific Research | 577 |
| Psychology | 143 |
| Role Playing | 233 |
| Other | 1819 |

while **4.8** constraints per instruction are generated using rule-based heuristics (rule-based constraints). Figure 4 shows the constraint density (i.e., constraints per instruction) distribution of RECAST-30K. The results suggest that RECAST-30K achieves a high level of constraint density, underscoring the dataset's broad coverage of constraints. This is further evidenced by the following statistics:

- **36.4%** (10,884 samples) of instructions contain **15 or more constraints**;
- **78.8%** (23,580 samples) of instructions contain **10 or more constraints**;
- The instruction with the highest complexity contains **over 30 constraints**.

As discussed in Appendix D.6, we also provide detailed statistics under different setups in Figures 5, 6, and 7, showing distribution of datasets' variants with different maximum constraint limits (5, 10, and 15).

This level of constraint density pushes the limits of current models' ability to handle compositional and multi-faceted constraints, representing a significant leap beyond prior instruction datasets.

### B.2.2 CONSTRAINT TYPES

In this section, we provide an in-depth analysis on the distribution of constraint type in RECAST-30K and its variants.

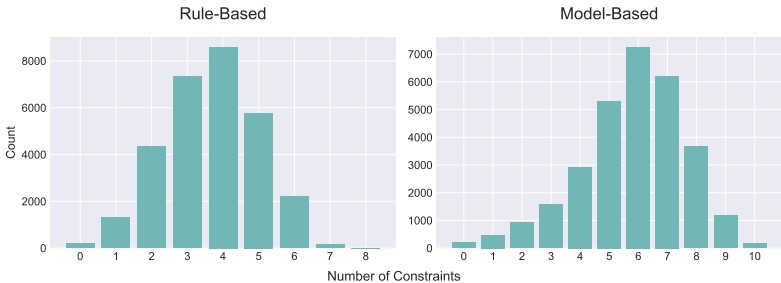

Figure 6: Constraint Count Distribution of Train Set with 10 Constraints.

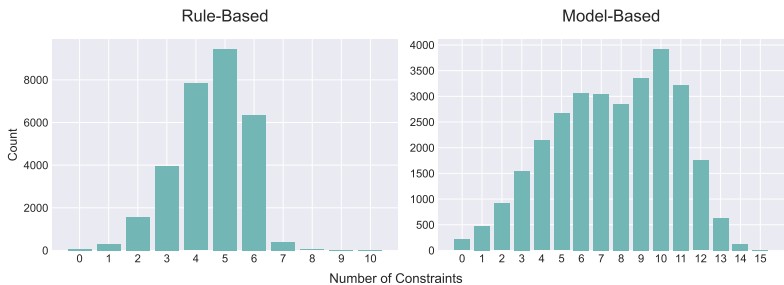

Figure 7: Constraint Count Distribution of Train Set with 15 Constraints.

**Visualizing the Distribution of Constraint Types**    To understand how constraint types are distributed in RECAST-30K, we visualize their distribution in Figure 8. The results suggest that RECAST-30K contains a highly diverse set of constraints that closely reflect real-world user instructions. Similarly, we provide results of the RECAST-30K variants with reduced sets of constraints. Figures 9, 10, and 11 visualize the constraint type distributions when the number of constraints is limited to 5, 10, and 15, respectively.

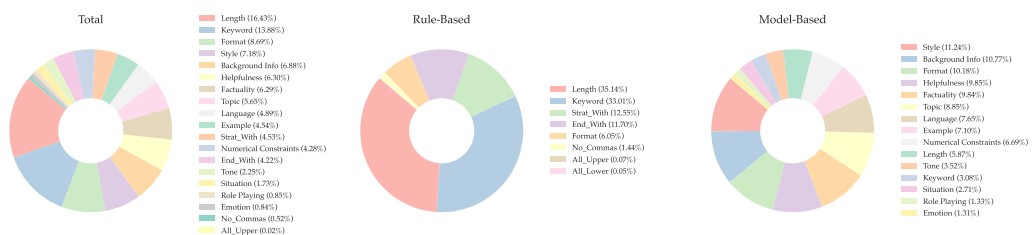

Figure 8: Constraint Type Distribution of RECAST-30K.

**Analysis of the Most and Least Common Constraint Types**    To provide further insights into the constraint type distribution in RECAST-30K, we examine which constraint types occur most and least frequently. The results are shown in Table 6. For model-based constraints, the three most frequently applied types are Style, Background Information, and Format, whereas the least frequent types are Emotion, Role Playing, and Situation. These model-based constraints span a broad range of semantic and stylistic dimensions. For rule-based constraints, the most common types are Length, Keyword, and Start With, while the least common types are All Lowercase, All Uppercase, and No Commas.

This complementary distribution reflects the design goal of balancing semantic richness (via model-based constraints) with the granularity of constraints (via rule-based constraints). The rule-based constraints effectively supplement the LLM-generated ones by covering fine-grained and often under-represented aspects of output control, thus improving both the coverage and accuracy of constraint specification.

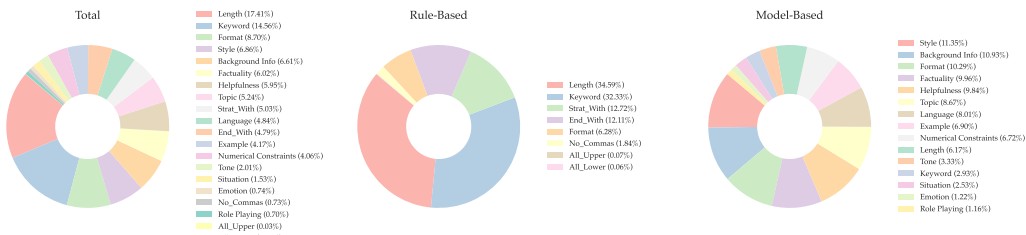

Figure 9: Constraint Type Distribution of Train Set with 5 Constraints.

Table 6: Most and Least Frequent Constraint Types

| Construction Method | Most Frequent Type (Top 3) | Least Frequent Type (Top 3) |
|---|---|---|
| Model-Based | Style, Background Information, Format | Emotion, Role Playing, Situation |
| Rule-Based | Length, Keyword, Start With | All Lowercase, All Uppercase, No Commas |

### B.2.3 DISCUSSION ON REALISM AND DIVERSITY

One of the key advantages of RECAST is that the constraints were not arbitrarily imposed but instead derived from naturally occurring responses. This ensures that the constraints are realistically achievable and semantically coherent with the task context. In addition to their authenticity, the constraints in RECAST-30K exhibit remarkable density, with many instructions containing a large number of constraints, and show high diversity across different constraint types. As a result, RECAST-30K serves as a rich and challenging training resource for enhancing models' ability to follow complex instructions in settings that closely resemble real-world usage, where users often issue requests involving multiple, varied, and fine-grained constraints.

### B.3 COST ANALYSIS OF CONSTRUCTING RECAST-30K

To provide transparency on the resource requirements of building our dataset, we report the estimated costs incurred at each stage of the RECAST pipeline. The total expenditure was approximately $175, broken down as shown in Table 7.

Table 7: Estimated cost of constructing RECAST-30K across different pipeline stages.

| Pipeline Stage | Estimated Cost (USD) |
|---|---|
| Rule-based Constraints Construction | $0 (no API calls required) |
| Model-based Constraints Construction | $70 |
| Instruction Enhancement | $85 |
| Majority Voting | $20 |
| **Total** | **$175** |

These figures demonstrate that RECAST-30K can be constructed at relatively low cost, underscoring the scalability and practicality of our pipeline for broader adoption in future instruction-following research.

### B.4 RECAST-TEST

To facilitate rigorous evaluation of complex instruction following capabilities across varying difficulty levels, we constructed **RECAST-Test** through a structured, progressive constraint selection approach. We began by randomly sampling 500 data points from RECAST-30K that contained at least 15 distinct constraints per

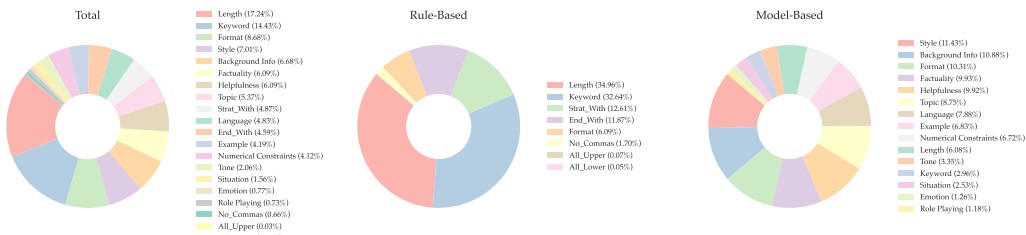

Figure 10: Constraint Type Distribution of Train Set with 10 Constraints.

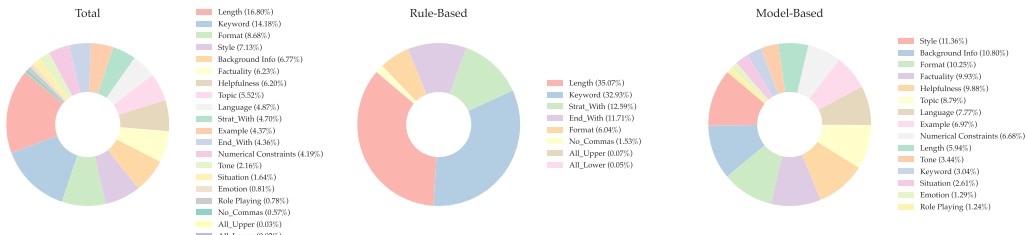

Figure 11: Constraint Type Distribution of Train Set with 15 Constraints.

instruction, establishing a robust foundation of constraint-rich examples. This metadata served as the basis for our hierarchical benchmark construction.

The key innovation in RECAST-Test lies in its carefully calibrated difficulty progression. We structured the benchmark into four distinct complexity tiers by implementing a constraint-nesting methodology:

- **Level 1 (Basic)**: Each instruction incorporates 5 constraints, providing a baseline assessment of fundamental instruction-following capability.

- **Level 2 (Intermediate)**: Instructions contain 10 constraints, including all 5 constraints from Level 1 plus 5 additional requirements.

- **Level 3 (Advanced)**: Instructions feature 15 constraints, encompassing all constraints from Level 2 with 5 additional requirements.

- **Level 4 (Comprehensive)**: Instructions include all available constraints for each sample, representing the maximum complexity level.

This nested constraint design ensures consistent difficulty progression while maintaining semantic coherence across levels. By preserving all constraints from previous levels, we enable direct performance comparisons as complexity increases.

For each difficulty tier, we employed the instruction enhancement pipeline from RECAST to seamlessly integrate the selected constraints into coherent prompts. This process involved using majority voting among multiple LLMs to generate linguistically natural instructions that incorporate all specified constraints without compromising readability or coherence. The resulting benchmark comprises four progressively challenging evaluation sets, each containing the same underlying tasks but with increasing constraint complexity, allowing for controlled assessment of instruction-following capabilities under varying levels of constraint demands.

**Visualizing the Distribution of Constraint Density**  To provide a comprehensive analysis of the constraint density distribution within the RECAST-Test, we conducted an in-depth examination of the constraint density across different subsets of RECAST-Test. Figures 12, 13, 14, and 15 present the constraint density distributions of the full RECAST-Test and its variants with a maximum constraints limit of 5, 10, 15, respectively. These results collectively demonstrate the richness of constraint in the RECAST-Test, which is crucial for evaluating the effectiveness of models' ability to follow complex instructions. The detailed analysis of constraint density distribution across different subsets provides valuable insights into the dataset's structure and utility.

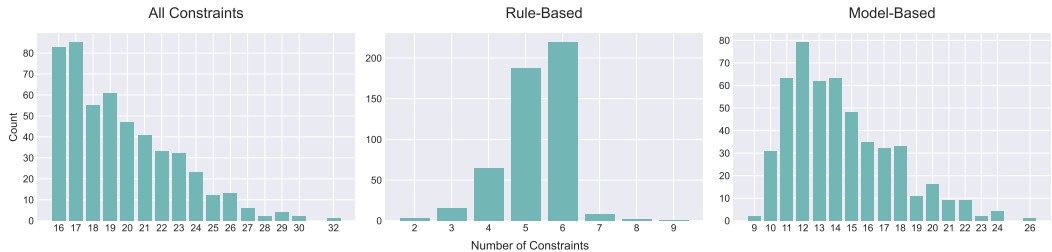

Figure 12: Constraint Density Distribution of RECAST-Test with All Constraints

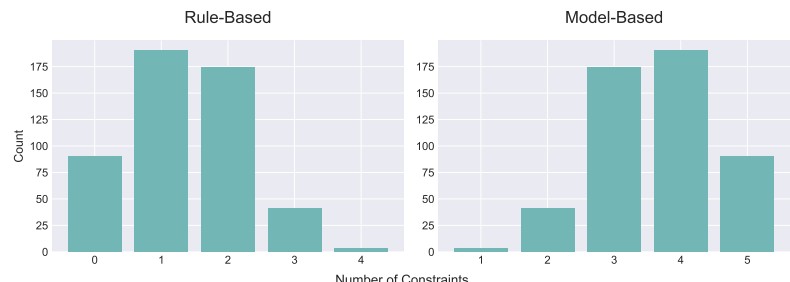

Figure 13: Constraint Density Distribution of RECAST-Test with 5 Constraints.

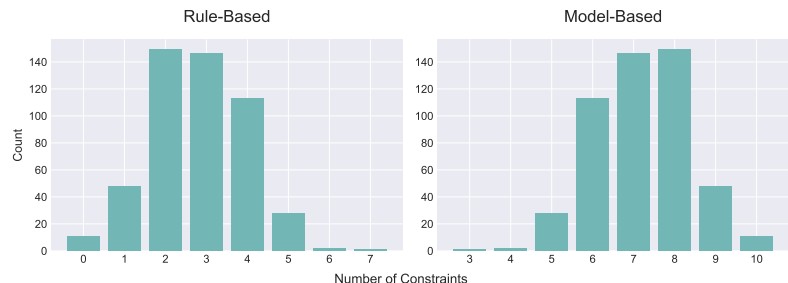

Figure 14: Constraint Density Distribution of RECAST-Test with 10 Constraints.

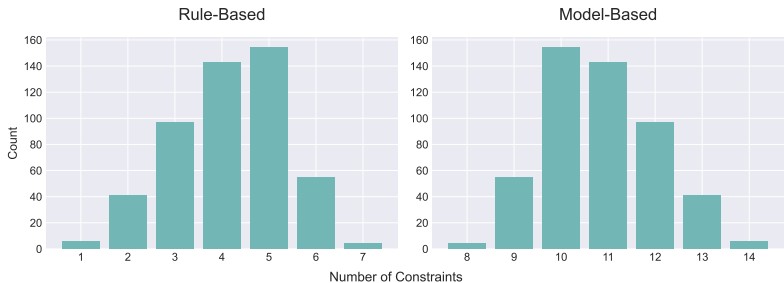

Figure 15: Constraint Density Distribution of RECAST-Test with 15 Constraints.

**Visualizing the Distribution of Constraint Types** Similarly, Figures 16, 17, 18, 19 illustrates the constraint type distributions of the full RECAST-Test with all constraints included and its variants. These results demonstrate that RECAST-Test covers a highly diverse range of constraint types, effectively capturing the complexity and variability found in real-world instructions. The detailed breakdown of constraint type

distributions across different variants offers valuable insights into the dataset's structural diversity and practical utility.

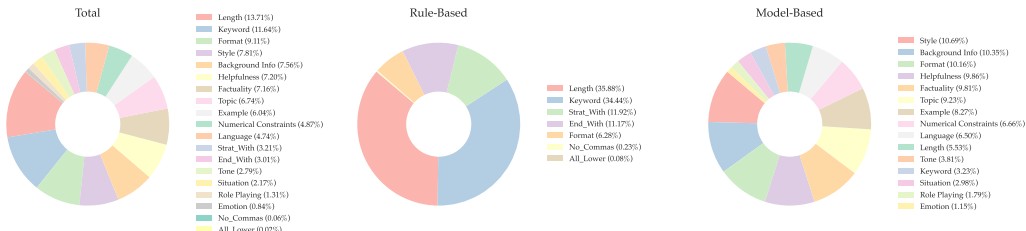

Figure 16: Constraint Type Distribution of RECAST-Test with All Constraints.

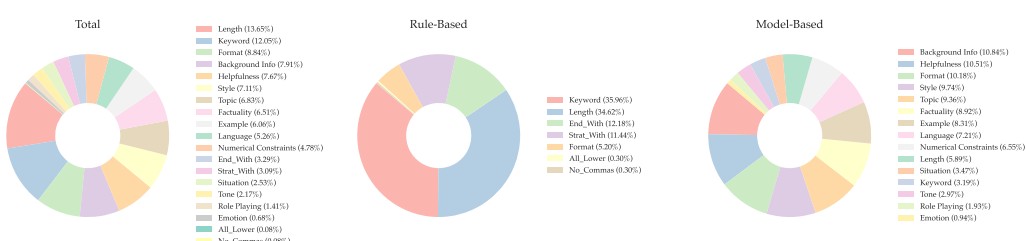

Figure 17: Constraint Type Distribution of RECAST-Test with 5 Constraints.

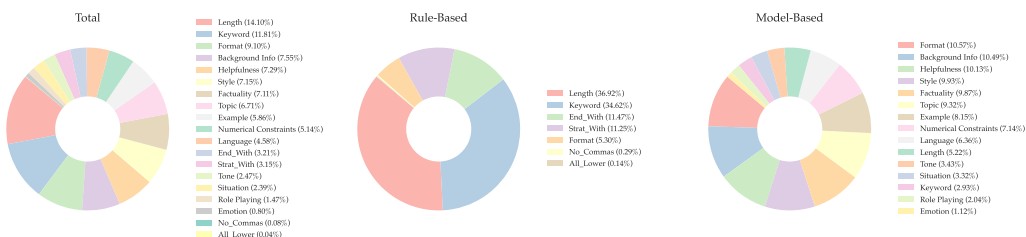

Figure 18: Constraint Type Distribution of RECAST-Test with 10 Constraints.

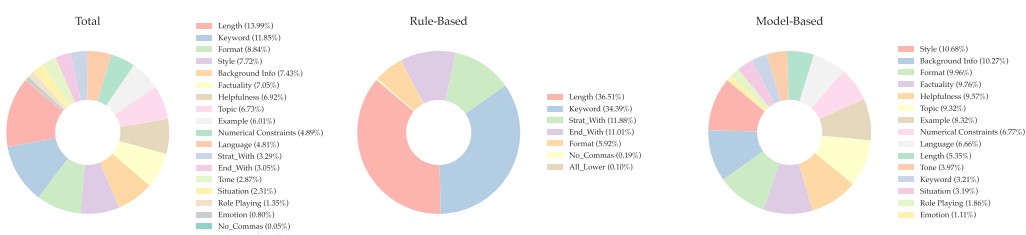

Figure 19: Constraint Type Distribution of RECAST-Test with 15 Constraints.

# C    HUMAN EVALUATION

In this study, three stages of human evaluation were conducted during the data generation process: constraint filtering, optimal instruction selection, and optimal response selection. The evaluation standards and results for each stage are as follows.

## C.1 Evaluation for Constraint Filtering

During the constraint filtering process, GPT-4o[1] was used to automatically filter out constraints that were not adhered to by the corresponding responses. After the filtering process, we obtained **355,751** high-quality constraints. To ensure the accuracy of the filtering process, we conducted human evaluation to verify whether the filtered constraints were truly not followed. The evaluation results indicated that there was a high level of consistency, reaching **90.1%**, between the judgments of human experts and the model's determinations. These results are summarized in Table 8.

## C.2 Evaluation for Optimal Instruction Selection

To select the optimal instruction, we employed multiple models to rewrite the original instructions. Specifically, we utilized four models—GPT-4o, Qwen-Plus[2], DeepSeek-V3[3], and Doubao-1.5-Pro-32k-250115[4]—to generate alternative versions of the instructions. These models ranked the instructions according to the criteria outlined below. The optimal instruction was then determined using the Borda algorithm. To ensure the accuracy of the model-selected instructions, we conducted a human evaluation where experts selected the optimal instructions based on the same criteria. We then compared the consistency between the model-selected and human-selected instructions. Both LLM and human evaluations were based on the following two criteria:

- **Clarity of Requirements:** Whether the instruction is clear and actionable.
- **Language Fluency:** Whether the instruction is grammatically correct and easy to understand.

The human evaluation results showed that the best instruction selected by LLM and the best instruction selected by human experts were consistent **94.6%** of the time. These results are also summarized in Table 8.

## C.3 Evaluation for Optimal Response Selection

To obtain high-quality responses, we employed four models—GPT-4o, Qwen-Plus, DeepSeek-V3, and Doubao-1.5-Pro-32k-250115—to generate responses to the complex instructions we constructed. To select the optimal response, these four LLMs ranked the responses based on the following five criteria:

- **Instruction Adherence:** Whether the response follows the constraints of the instruction.
- **Helpfulness:** Whether the response fully addresses the user's request.
- **Accuracy:** Whether the information is correct and reliable.
- **Clarity:** Whether the response is well-structured and easy to understand.
- **Conciseness:** Whether the response is concise and free of unnecessary details.

After ranking the four responses using LLMs, we applied the Borda algorithm to select the optimal response for inclusion in our dataset. To verify the accuracy of the LLM-selected optimal response, we had human experts select the optimal response based on the same criteria and then compared the consistency between the LLM-selected and human-selected optimal responses. The results indicated that the optimal response selected by the LLM was consistent with the optimal response selected by human experts **95.7%** of the time. These results are summarized in Table 8.

## C.4 Impact of Model-based Validator Noise

The human evaluations in Appendix C.1 and Table 8 show that our model-based validators achieve over **90%** agreement with human experts, indicating that the verification signal is generally reliable. Beyond these aggregate agreement numbers, we also argue that RLVC is intrinsically robust to residual validator errors.

First, the RLVC reward for each trajectory is computed as the mean satisfaction rate over a dense set of constraints (typically $> 10$ per instruction). This averaging naturally attenuates the influence of *uncorrelated* false positives and false negatives: occasional misjudgments on individual constraints are diluted when aggregated across many constraints, so the overall reward signal remains stable. Second, policy-gradient methods such as GRPO optimize expected returns over large batches and are known to tolerate moderate levels of reward noise.

This robustness is also consistent with recent RLHF findings, which show that reward models with only *moderate* agreement with human preferences (often around 60–70%) are already sufficient to drive effective alignment

---

[1] https://platform.openai.com/docs/models/gpt-4o
[2] https://bailian.console.aliyun.com
[3] https://platform.deepseek.com/usage
[4] https://console.volcengine.com

Table 8: Summary of Human Evaluation Results

| Evaluation Item | Standards | Consistency (%) |
|---|---|---|
| Constraint Filtering | **Filtering Accuracy:** Whether the response indeed violates the filtered constraints. | 90.1 |
| Instruction Voting | **Clarity of Requirements:** Whether the instruction is unambiguous and directly actionable. | 94.6 |
| | **Language Fluency:** Whether the instruction is grammatically correct and easy to understand. | |
| Response Voting | **Instruction Adherence:** Whether the response follows the given instruction. | 95.7 |
| | **Helpfulness:** Whether the response fully addresses the user's request. | |
| | **Accuracy:** Whether the information is correct and reliable | |
| | **Clarity:** Whether the response is well-structured and easy to understand. | |
| | **Conciseness:** Whether the response is efficient and avoids unnecessary details. | |

and improve downstream behavior(Chen et al., 2024; Tyen et al., 2024). In contrast, our model-based validators provide a substantially higher-fidelity signal (over 90% human agreement). If the noise level were catastrophic, we would expect training instabilities or systematic degradation (e.g., learning spurious patterns or reward hacking). Instead, we observe steady improvements in model-based satisfaction rates (MSR) and overall success rates (OSR) in our experiments, suggesting that RLVC successfully extracts the underlying semantics of the constraints despite the presence of minor verification errors.

Moreover, our model-based validators only make *binary* (satisfied / not satisfied) decisions rather than assigning graded scores, which reduces the difficulty of the LLM-as-a-judge task and improves the stability and interpretability of the verification signal. This design choice is aligned with a growing body of work that uses LLMs, often via rubrics or checklists, to evaluate whether responses satisfy structured criteria, and reports that such LLM-based evaluations can be robust and practical in diverse text-generation settings(Cook et al., 2024; Lee et al., 2025). Together with our human–LLM agreement analysis, these results further support that the residual noise in our model-based validators is moderate and that RLVC can safely leverage such validators as reward sources.

## D    EXPERIMENTAL DETAILS

### D.1    BASELINE

We conduct comparisons against eight representative datasets specifically curated to improve the instruction-following capabilities of language models, namely:

- **Conifer** (Sun et al., 2024): Constructs a GPT-4-refined dataset with hierarchical constraint instructions and employs curriculum tuning to enhance complex instruction-following.
- **Crab** (Qi et al., 2024): Enhances LLMs by back-translating model responses into constraint-rich instructions using a stronger teacher model.
- **I-SHEEP** (Liang et al., 2024): Aligns LLMs from scratch via iterative self-generated instruction–response tuning without external supervision.
- **MUFFIN** (Lou et al., 2023): Diversifies instruction data by scaling input facets per task, yielding more robust multi-faceted alignment.
- **ShareGPT**[5]: Provides real user–assistant dialogues as a large-scale resource for training conversational instruction-following.

---

[5]`https://huggingface.co/datasets/anon8231489123/ShareGPT_Vicuna_unfiltered`

- **Evol-Instruct** (Xu et al., 2024): Generates complex instructions through iterative rewriting, producing progressively harder instruction–response pairs.

- **Suri** (Pham et al., 2024): Back-translates long-form texts into multi-constraint prompts, improving structured generation over extended contexts.

- **Tülu 3 Persona IF** (Lambert et al., 2024): Synthesizes constraint-following data from diverse persona prompts to enhance controllable instruction tuning.

## D.2 GENERAL CAPABILITY EVALUATION BENCHMARK

We employ four diverse benchmarks spanning different aspects of language understanding and reasoning:

- **FollowBench** (Jiang et al., 2024b) comprehensively evaluates instruction-following capability through five types of fine-grained constraints (Content, Situation, Style, Format, and Example) using a multi-level mechanism that incrementally adds constraints to initial instructions. This benchmark systematically measures models' ability to satisfy multiple constraints simultaneously, with difficulty levels ranging from simple single-constraint instructions to complex multi-constraint scenarios. The evaluation employs metrics including Hard Satisfaction Rate (HSR) for complete constraint satisfaction and Soft Satisfaction Rate (SSR) for partial compliance.

- **IFEval** (Zhou et al., 2023) assesses models' ability to follow explicit formatting instructions through strict, verifiable metrics rather than subjective content evaluation. The benchmark focuses on precise adherence to specific requirements such as keyword inclusion, format specifications, and structural constraints. This allows for rigorous, code-based evaluation that eliminates ambiguity in measuring instruction compliance.

- **GPQA** (Rein et al., 2023) features expert-crafted questions from PhD-level domain specialists in biology, physics, and chemistry designed to be challenging for non-experts yet accessible to specialists. The dataset undergoes rigorous validation to ensure both factual accuracy and appropriate difficulty levels. Access restrictions and gating mechanisms protect against data contamination, maintaining the benchmark's integrity for evaluating advanced scientific knowledge.

- **MuSR** (Sprague et al., 2024) tests models' ability to integrate reasoning with long-context understanding through algorithmically generated complex problems averaging 1,000 words in length. The benchmark includes murder mysteries, object placement puzzles, and team allocation optimization tasks that require maintaining coherent reasoning across extended contexts. Few models achieve above-random performance, making it an effective discriminator of advanced reasoning capabilities.

## D.3 DETAILS OF DATA GENERATION

We utilize Tülu 3 Persona IF[6] as seed data, which contains instruction following tasks in various real-world scenarios. We employed Large Language Models (LLMs) extensively throughout the data generation pipeline, encompassing constraint generation, constraint filtering, instruction rewriting, optimal instruction selection, response regeneration, and optimal response selection. Specifically, for constraint generation and filtering, we utilized the API service of `deepseek-V3`. In the instruction rewriting phase, we leveraged the API services of four distinct models: `GPT-4o`, `Qwen-Plus`, `DeepSeek-V3`, and `Doubao-1.5-Pro-32k-250115`. These models generated four alternative versions of complex instructions. Subsequently, the selection of the optimal instruction involved ranking these four versions using the same set of four models. The top-ranked instruction was then chosen for the subsequent response regeneration step. Similar to instruction rewriting, we again invoked the API services of `GPT-4o`, `Qwen-Plus`, `DeepSeek-V3`, and `Doubao-1.5-Pro-32k-250115` to generate four candidate responses based on the selected optimal instruction. Finally, these four responses were ranked using the same four models, and the highest-ranked response was adopted as the final answer within our dataset.

During all API calls, we maintained consistent parameter settings: `temperature`=0, `top_p`=1, and `n`=1. We deliberately set the `temperature` to 0 to prioritize the accuracy of the generated data, which is paramount for complex instruction-following tasks. While the diversity within our dataset stems from the rich semantics inherent in real-world responses, maintaining accuracy during constraint generation, instruction rewriting, response regeneration, and the selection processes for both instructions and responses was our primary concern. Therefore, a temperature of 0 was chosen to maximize the determinism and factual correctness of the LLM outputs at each stage of the data generation process.

---

[6]https://huggingface.co/datasets/allenai/tulu-3-sft-personas-instruction-following

### D.4 DETAILS OF SFT

This section details our model training configuration based on the LlamaFactory (Zheng et al., 2024) framework. We employed Supervised Fine-Tuning (SFT) with a maximum sequence length of 4096 tokens. For optimization, we used a linear learning rate scheduler with a peak learning rate of 2.0e-5, 3% warmup ratio, and trained for 3 epochs. We configured training with a per-device batch size of 4 and gradient accumulation steps of 32, resulting in an effective batch size of 128. To optimize training efficiency, we utilized DeepSpeed with ZeRO stage 3 optimization and bfloat16 mixed precision. All experiments were conducted on a cluster of 8 NVIDIA H800 GPUs.

### D.5 DETAILS OF RLVC

**Policy Optimization**    We employ Group Relative Policy Optimization (GRPO) (Shao et al., 2024) as our policy learning algorithm. GRPO generates multiple candidate responses for the same instruction and computes advantage estimates through intra-group comparisons, allowing the model to learn the relative quality differences among various responses to the same instruction.

Concretely, for every input instruction $x$, the current policy $\pi_\theta$ samples $G$ candidate responses $\{y_i\}_{i=1}^G$ and obtains a scalar reward $r_i$ for each. We then compute the group mean $\mu$ and standard deviation $\sigma$, and derive a standardised advantage $\hat{A}_i$, which measures how well each response performs relative to its peers.

During policy updates, GRPO maximises the following objective:

$$
\mathcal{J}_{\text{GRPO}}(\theta) = \mathbb{E}_{x \sim P(X),\ \{y_i\}_{i=1}^G \sim \pi_{\theta_{\text{old}}}(Y|x)} \Bigg[
$$
$$
\frac{1}{G} \sum_{i=1}^G \frac{1}{|y_i|} \sum_{t=1}^{|y_i|} \left\{ \min \left( r_{i,t} \hat{A}_{i,t},\ \text{clip}\left(r_{i,t},\ 1-\varepsilon,\ 1+\varepsilon\right) \hat{A}_{i,t} \right) - \beta\, \mathbb{D}_{\text{KL}}\left[\pi_\theta \,\|\, \pi_{\text{ref}}\right] \right\} \Bigg]
\tag{4}
$$

where:

$$
\mu = \frac{1}{G} \sum_{i=1}^G R_i,\ \sigma = \sqrt{\frac{1}{G} \sum_{i=1}^G (R_i - \mu)^2},\ \hat{A}_i = \frac{R_i - \mu}{\sigma}, r_{i,t}(\theta) = \frac{\pi_\theta(y_{i,t} \mid x, y_{i,<t})}{\pi_{\theta_{\text{old}}}(y_{i,t} \mid x, y_{i,<t})}
\tag{5}
$$

By combining constraint-specific rewards with GRPO's comparative learning approach, our framework provides the model with clear guidance on how to improve its constraint satisfaction across diverse instruction types.

**Implementation Details**    For reinforcement learning, we implemented our RLVC approach based on the VeRL (Sheng et al., 2025) framework with Group Relative Policy Optimization (GRPO). Our configuration used a learning rate of 1e-6 with maximum sequence lengths of 1024 tokens for both prompts and responses. We employed a batch size of 512 with PPO mini-batches of 128 and micro-batches of 16 per GPU. For optimization stability, we incorporated KL divergence regularization with a coefficient of 0.001 using the low-variance KL implementation, while enabling gradient checkpointing for memory efficiency. The rollout process utilized tensor model parallelism with a size of 2 and vLLM for acceleration, generating 16 candidate responses per prompt with 60% GPU memory utilization. We integrated our custom constraint-specific reward functions through the thread-based reward manager with asynchronous reward calculation. All experiments were conducted on 8 GPUs for 200 optimization steps.

**Training Dynamics**    To better understand the optimization behaviour of RLVC, we track the evolution of both the model-based and rule-based rewards, together with their corresponding satisfaction rates, during the early phase of RL training. Table 9 reports the recorded values at steps 0, 50, 100, 150 and 200.

The observed dynamics can be characterised in three phases. First, an *early surge* (0–100 steps): the model rapidly increases its model-based reward as it captures a set of relatively easy model-based constraints, accompanied by gradual improvements in rule-based satisfaction. Second, a *slow-down phase* (after ~100 steps): once these low-hanging model-based objectives are largely addressed, optimization attention shifts toward more difficult, often rule-based objectives, causing the model-based reward to plateau or decline slightly, while rule-based metrics begin to improve more noticeably. Third, the *overall trend*: despite the plateau in the model-based reward, both MSR and RSR show mild upward trajectories, indicating that the model is improving its aggregated ability to satisfy an increasingly challenging mixture of constraints.

Notably, the scale of improvements differs across metric types: model-based reward increases modestly by about 8% relative (from 0.411 to 0.445 at peak), whereas rule-based reward—though starting at a much lower absolute value—doubles in magnitude and its satisfaction rate (RSR) grows from 4.8% to 10.9%. This contrast highlights

Table 9: Reward metrics and constraint satisfaction rates during the first 200 RLVC optimization steps.

| Step | 0 | 50 | 100 | 150 | 200 |
|---|---|---|---|---|---|
| Model-based reward | 0.411 | 0.428 | 0.445 | 0.431 | 0.415 |
| MSR (%) | 54.0 | 56.5 | 57.5 | 56.0 | 55.5 |
| Rule-based reward | 0.011 | 0.022 | 0.016 | 0.030 | 0.018 |
| RSR (%) | 4.8 | 5.5 | 5.4 | 5.8 | 10.9 |

the higher difficulty but also the meaningful progress in handling rule-based objectives. These dynamics suggest that RLVC first exploits readily learnable, model-evaluable constraints before reallocating capacity to harder objectives; consequently, monitoring both per-constraint-type rewards and aggregated metrics is important for diagnosing progress and for informing choices such as reward balancing, curriculum scheduling, or extended training.

### D.6 ABLATIONS AND ANALYSIS

**Impact of constraint type during complex instruction learning.** To understand the contribution of different constraint types, we conducted experiments using RECAST with only model-based or only rule-based constraints, the results are shown in Figure 21. Results indicate that constraint type specialization affects performance metrics. Models trained with only model-based constraints demonstrate reduced performance on RSR, while maintaining competitive MSR scores. Conversely, models trained with only rule-based constraints exhibit decreased model-based constraint satisfaction capabilities, particularly evident in higher difficulty levels. This specialization effect highlights the importance of diverse constraint exposure during training. These findings underscore the necessity of incorporating both kinds of constraint types during model training to ensure robust performance across different complexity levels.

**Generalization across constraint types.** To further assess whether RECAST-trained models can handle constraint types beyond those explicitly used during training, we conduct a held-out–type analysis using Llama-3.1-8B. Concretely, we compare three training configurations: (i) *RECAST-30K (all types)*, which uses both model-based and rule-based constraints; (ii) *Only Model-based Constraints*, where the model is trained solely on model-based constraints but evaluated on both MSR and RSR; and (iii) *Only Rule-based Constraints*, where the model is trained solely on rule-based constraints but again evaluated on both MSR and RSR. We additionally include a strong instruction-following baseline (Tülu 3 Persona IF) that does not use RECAST-30K for comparison. The numerical results are shown in Figure 21.

The results reveal two key trends. First, a model trained only on model-based constraints still achieves competitive RSR and clearly outperforms the baseline on rule-based constraints. Symmetrically, a model trained only on rule-based constraints attains strong MSR and surpasses the baseline on model-based constraints. Second, using all constraint types yields the best overall performance, but removing one type does not cause a catastrophic drop on the other type. Together, these findings suggest that RECAST training does not merely overfit to specific constraint templates; instead, it encourages the model to acquire more generalizable instruction-following behavior that transfers across different families of constraints, providing partial evidence toward robustness to novel constraint types.

**Impact of constraint number during complex instruction learning.** To investigate how constraint quantity affects instruction-following capabilities, we trained variants of RECAST-30K-SFT with different maximum constraint limits (5, 10, and 15) and evaluated their performance across all difficulty levels, the result can be found in Figure 20. Results demonstrate a clear alignment between training constraint quantity and evaluation performance at corresponding difficulty levels. Notably, the complete RECAST-30K-SFT model maintains competitive performance across all difficulty levels despite not being specialized for any particular constraint quantity. This balanced performance profile suggests that exposure to diverse constraint quantities during training enables effective generalization across varying levels of instruction complexity.

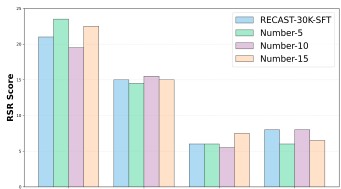

Figure 20: Impact of constraint number during complex instruction learning.

**Impact of specific components of RECAST.** To isolate component contributions, we evaluated RECAST variants with specific elements removed(Table 10). Complete

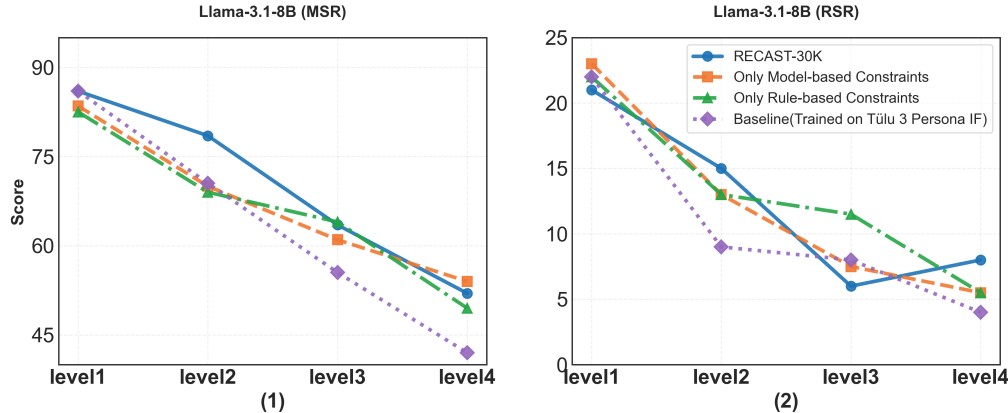

Figure 21: Impact of constraint type during complex instruction learning.

Table 10: Average results for Llama-3.1-8B and Qwen2.5-7B across different RECAST components.

| Configuration | Average | |
|---|---|---|
| | Llama-3.1-8B | Qwen2.5-7B |
| **RECAST-30K** | **30.54** | **31.25** |
| **Response-Only Enhancement** | 27.00 | 25.54 |
| **Instruction-Only Enhancement** | 18.33 | 13.88 |
| **Without RECAST** | 26.96 | 26.83 |

RECAST-30k consistently outperforms all ablated configurations. Instruction-Only Enhancement exhibits the most significant performance degradation (12.21% and 17.37% decreases for Llama and Qwen), revealing that constraint-augmented responses without corresponding response modifications create instruction-response inconsistencies that confuse the model, substantially hampering performance. Response-Only Enhancement performs comparably to Without RECAST, suggesting that constraint specification in instructions drives most performance gains. These results confirm that RECAST's complete instruction-response synthesis pipeline is essential for optimal constraint satisfaction capabilities.

**Binary evaluation for model-based constraints.** Some high-level model-based constraints, such as *Helpfulness*, may in principle support finer-grained distinctions than a simple binary satisfied / not satisfied label: two responses can both be helpful while still differing noticeably in quality. A learned reward model that produces graded scores could better capture such nuances and enable more fine-grained optimization. In this work, however, our primary focus is on whether a model can simultaneously satisfy multiple constraints within a single prompt, i.e., the *breadth* of constraint satisfaction across a complex instruction. To keep this focus clear, we deliberately relax the requirement on the *depth* of satisfaction for any single high-level constraint and adopt a binary criterion at the per-constraint level. In other words, there is an inherent trade-off between modeling the breadth of multi-constraint satisfaction and capturing very fine-grained differences within each individual constraint, and our design in this paper prioritizes the former. Exploring combinations of RLVC with more fine-grained, scalar reward models for certain model-based constraints is an interesting direction for future work.

**Applicability of RLVC to reasoning-style base models.** RLVC is conceptually agnostic to the choice of base model: it only requires that, given an instruction with verifiable constraints, the model produces a final response that can be checked by our validators. In this sense, the framework applies equally well to reasoning-style base models and to non-reasoning models such as Qwen-2.5-7B and Llama-3.1-8B used in our experiments. For many reasoning models, the interface consists of a prompt, an internal (possibly explicit) "thinking" trace, and a final answer. Existing reasoning-oriented training pipelines typically supervise the *final response* while allowing the model to freely choose its internal reasoning. RLVC fits naturally into this paradigm: our constraint verifiers operate on the final response, and the resulting reward can be used to train reasoning models in exactly the same way, without requiring access to or supervision over their internal thinking process. Looking ahead, one could also extend RLVC to explicitly supervise the thinking process itself, for example by augmenting data with "how to satisfy each constraint" reasoning traces and defining verifiable constraints over

these traces. We view such tighter coupling between verifiable constraints and chain-of-thought supervision as promising future work.

# E    CONSTRAINT TAXONOMY

Prior to generating constraints, we constructed a comprehensive constraint taxonomy composed of two categories: **rule-based constraints** and **model-based constraints**. The constraint categories include 19 types, with definitions and representative examples shown in Table 11 and Table 12. These types serve as the basis for generating diverse and realistic constraints.

## E.1    RULE-BASED CONSTRAINTS

**Rule-based constraints** are objective requirements that can be verified using deterministic methods, such as Python functions. These constraints are structural, lexical, or quantitative in nature, such as paragraph count, keyword inclusion, and word limits. To generate these constraints, we implemented nine rule-based extractors that identify verifiable properties from text. These extractors analyze syntactic patterns, keyword frequency, numerical values, and other measurable elements to produce concrete constraint instances.

Table 11: Rule-based Constraint Types

| Constraint Type | Definition | Examples |
|---|---|---|
| Length | The response is required to adhere to a specific length. | Respond in roughly 100 to 200 words. |
| Format | The response is required to follow a particular format. | Format ingredients as a bulleted list. |
| Language | The response is required to be written in a specific language or to use uppercase/lowercase. | Write in English. |
| Keyword | The response is required to include specific words or phrases. | Incorporate 1 "The Civil Rights Act of 1964" into your answer. |
| Start With | The response is required to contain numerical patterns. | Begin your response with the word "dear". |
| End With | The response is required to start with specific word or phrase. | End your response with the word "HARMONY". |
| All Upper | The response is required to be in all capital letters. | Type everything in CAPS LOCK. |
| All Lower | The responses are required to be in all lowercase letters. | Use only small letters in your answer. |
| No Commas | The response is required to contain numerical patterns. | Avoid commas entirely in your answer. |

## E.2    MODEL-BASED CONSTRAINTS

**Model-based constraints** are subjective and require semantic interpretation or qualitative judgment. These involve higher-level aspects such as style, tone, emotion, and content relevance. We developed a taxonomy of 10 constraint types encompassing semantic fidelity, stylistic nuance, tonal variation, and domain-specific content requirements. These were used to guide the LLM in generating rich and context-sensitive constraint instances for each response.

Table 12: Model-based Constraint Types

| Constraint Type | Definition | Examples |
| --- | --- | --- |
| Tone | The response is required to use a specified tone. | Maintain a formal and academic tone. |
| Emotion | The response is required to convey a certain emotion. | Convey nostalgia and hopeful anticipation. |
| Style | The response is required to reflect a particular style. | Write in a poetic style with a consistent AABB rhyme scheme. |
| Factuality | The response is required to focus on verifiable facts or imaginative content. | Include only medically verified items. |
| Helpfulness | The response is required to provide useful information. | Provide actionable advice for emergency preparedness. |
| Example | The response is required to contain explicit examples. | Mention key testimonies from friends and financial experts. |
| Background Info | The response is required to be generated based on background information from a specific field. | Base the response on legal trial procedures and criminal case details |
| Role Playing | The response is required to simulate a specific role. | Adopt the role of a cancer researcher proposing a study. |
| Topic | The response is required to revolve around a particular subject or theme. | Focus exclusively on psychological effects of intense training on children. |
| Situation | The response is required to be generated based on a specific scenario or setting. | Frame the response as a public announcement for a hotel redesign. |

### E.3   TEMPLATES FOR RULE-BASED CONSTRAINT GENERATION

To generate rule-based constraints, we designed a set of eight rules and associated rule-based evaluation functions, which systematically assess responses within the seed dataset. When a response exhibited characteristics defined by our rules, we tailored a constraint for it. This constraint was not only precisely adhered by the response but also had corresponding functions for verification. To prevent the constraints constructed based on rules from being overly uniform, we designed multiple templates for each type of constraint. When generating specific constraints for a response, we randomly selected a template to ensure the diversity of the constraints. The templates for each type of constraint are as follows.

#### E.3.1   TEMPLATES FOR LENGTH CONSTRAINTS (WORD LEVEL)

The templates for word-level length constraints are summarized in Table 13. These templates cover three types of rules: Approximate, Below, and Range. Each rule type has multiple templates to ensure diversity.

Table 13: Templates for Word-level Length Constraints

| Rule | Description | Templates |
|------|-------------|-----------|
| Approximate | Whether the word count of the response is around some given number. | Use around {} words. |
| | | Limit your response to approximately {} words. |
| | | Aim for around {} words in your answer. |
| | | Keep your answer close to {} words. |
| | | Provide a detailed response of approximately {} words. |
| | | Your answer should be about {} words, plus or minus 20%. |
| | | Aim for approximately {} words. |
| | | Keep your answer around {} words. |
| Below | Whether the response contains fewer words than a specified maximum. | Keep the answer under {} words. |
| | | No more than {} words. |
| | | Do not exceed {} words. |
| | | Strictly limit the answer to {} words. |
| | | Stay within {} words. |
| | | {} words maximum. |
| | | Use fewer than {} words. |
| | | Cap your response at {} words. |
| | | Answer in {} words or less. |
| Range | Whether the response length falls within a specified word range. | Limit the response to {}–{} words. |
| | | Respond in roughly {} to {} words. |
| | | Target a response between {} and {} words. |
| | | Answer in approximately {}–{} words. |
| | | Keep the response between {} and {} words. |
| | | Aim for {} to {} words in your reply. |
| | | Limit your answer to a length of {}–{} words. |
| | | Adhere to a word count of {} to {}. |

### E.3.2 TEMPLATES FOR LENGTH CONSTRAINTS (SENTENCE LEVEL)

The templates for sentence-level length constraints are summarized in Table 14. These templates cover four types of rules: Exact, Approximate, Below, and Range. Each rule type has multiple templates to ensure diversity.

Table 14: Templates for Sentence-level Length Constraints

| Rule | Description | Templates |
|---|---|---|
| Exact | Whether the number of sentences in the response is exactly as specified. | Provide exactly {} sentences in your answer. |
| | | Use exactly {} sentences in your response. |
| | | Your response must contain exactly {} sentences. |
| | | Strictly use {} sentences in your answer. |
| | | Adhere to a limit of exactly {} sentences. |
| | | Structure your answer in exactly {} sentences. |
| | | Craft a {}-sentence response. |
| | | The answer shall comprise exactly {} sentences. |
| Approximate | Whether the number of sentences in the response is approximately equal to a specified number. | Aim for approximately {} sentences (±2). |
| | | Your answer should be around {} sentences, give or take a few. |
| | | Target around {} sentences in your response. |
| | | Keep your answer to roughly {} sentences. |
| | | Respond with approximately {} sentences. |
| | | The response should consist of approximately {} sentences. |
| Below | Whether the number of sentences in the response does not exceed a specified maximum. | Limit your response to {} sentences. |
| | | Use no more than {} sentences. |
| | | Do not exceed {} sentences in your response. |
| | | Cap your reply at {} sentences. |
| | | Stay within {} sentences. |
| | | Adhere to a maximum of {} sentences. |
| Range | Whether the number of sentences in the response falls within a specified sentence range. | Keep your answer to {}–{} sentences. |
| | | Respond in {} to {} sentences. |
| | | Provide a response of {}–{} sentences. |
| | | Aim for a response between {} and {} sentences. |
| | | Keep your reply within the range of {}–{} sentences. |
| | | The response should comprise {}–{} sentences. |
| | | Maintain a sentence count between {} and {}. |
| | | Provide an answer consisting of roughly {} to {} sentences. |

### E.3.3 TEMPLATES FOR OTHER CONSTRAINTS

The templates for other constraints are summarized in Table 15. These templates cover seven types of rules: Format, Keyword, Start With, End With, Uppercase, Lowercase and No Commas. Each rule type has multiple templates to ensure diversity.

Table 15: Templates for Other Constraints

| Rule | Description | Templates |
|------|-------------|-----------|
| **Format** | | |
| Format | Whether the response follows a specified output format such as JSON, list, paragraph, etc. | Respond in "{}" format. |
| | | Format your answer as valid "{}". |
| | | Provide the output strictly in "{}" format. |
| **Keywords** | | |
| Keyword | Whether the response includes a specified keyword a specified number of times. | Include the keywords "{}" {} times in your response. |
| | | Your response must feature "{}" {} times. |
| | | Use "{}" {} times when responding. |
| | | Ensure your answer contains {} "{}". |
| | | Incorporate {} "{}" into your answer. |
| **Position-Specific Strings** | | |
| Start With | Whether the response begins with a specified word or phrase. | Begin your response with the word "{}". |
| | | Start your answer with "{}". |
| | | Open your reply using "{}" as the first word. |
| End With | Whether the response ends with a specified word or phrase. | End your response with the word "{}". |
| | | Make sure the last word of your reply is "{}". |
| | | Your response must terminate with "{}". |
| **Letter Case** | | |
| Uppercase | Whether the response is written entirely in uppercase letters. | Write your entire response in UPPERCASE. |
| | | Use ONLY CAPITAL LETTERS in your answer. |
| | | Type everything in CAPS LOCK. |
| Lowercase | Whether the response is written entirely in lowercase letters. | Write your entire response in lowercase. |
| | | Use only small letters in your answer. |
| | | Avoid any capital letters in your reply. |
| **Punctuation** | | |
| No Commas | Whether the response excludes all commas from the output. | Do not use any commas in your response. |
| | | Avoid commas entirely in your answer. |
| | | Exclude all commas from your output. |

## F  MAIN PROMPTS

**Prompt Template for Constraint Generation via LLM**   Figure 22 illustrates the prompt template used to generate constraints from responses of seed dataset. Given a response and a list of constraint categories, the LLM is instructed to generate concrete, imperative-style constraints that can be attributed to each category. The template guides the LLM to return a structured dictionary, where each constraint type maps to one or more specific constraint instances written in command form (e.g., "use at least 100 words", "respond in formal tone"). This prompt plays a central role in generating a large number of diverse, multi-granular real-world constraints, enabling systematic conversion from abstract categories to enforceable constraints.

---

**Prompt Template for Constraint Generation via LLM**

Given An LLM response known to be generated under a prompt containing specific constraint categories and the constraint categories present in the original prompt.
Analyze the response to generate concrete imperative-style constraints for each constraint category below.

The LLM Response:
{LLM_RESPONSE}

The original prompt has the following constraints:
{CONSTRAINTS_EXPLANATION}

Output format:
Return a dictionary where:
 - Keys are constraint type
 - Values are lists of full imperative-form constraints such as such as {{"Constraint type1":["Concrete Constraint1","Concrete Constraint2"], "Constraint type2":["Concrete Constraint1","Concrete Constraint2","Concrete Constraint3"]}}
Don't output any additional text or explanations:

---

Figure 22: Prompt Template for Constraint Generation via LLM

**Prompt Template for Adding Constraints to Original Instructions**    Figure 23 presents a prompt template designed to augment original instructions with additional constraints in a fluent and semantically consistent manner. Given an initial instruction from the seed dataset and a dictionary of constraints, the LLM is prompted to modify the instruction by incorporating any missing constraints while preserving its original intent. This prompt is particularly useful for generating constraint-rich instruction datasets, as it ensures that generated instructions explicitly encode the intended execution requirements without altering their core meaning. The output is a single revised instruction that seamlessly integrates all constraint conditions.

**Prompt Template for Adding Constraints to Original Instructions**

Task: Constraint-Aware Instruction Expansion

You will be given:
1. A user instruction (to be given to another AI system)
2. A dictionary of constraints (requirements for how the instruction should be executed)

Your task is to MODIFY THE USER INSTRUCTION by:
1. Analyzing each constraint in the dictionary
2. For each constraint NOT already present in the user instruction:
   - Add it to the instruction in a natural, fluent way
   - Maintain the original meaning and flow
3. If a constraint is already satisfied by the instruction:
   - Leave it unchanged
4. Return ONLY the final modified instruction

CRUCIAL NOTES:
- You are NOT to apply the constraints to the instruction text itself
- You are ADDING REQUIREMENTS to the instruction
- Preserve the original instruction's intent and wording where possible

Example 1:
User Instruction: "Write a poem about the ocean"
Constraints: {{"length": "at least 20 lines", "style": "haiku"}}
Output: "Write a poem about the ocean that is at least 20 lines long and in haiku style"

Example 2:
User Instruction: "Summarize the text in 300 words"
Constraints: {{"length": "200-400 words", "tone": "professional"}}
Output: "Summarize the text in 300 words using a professional tone"

Now process the following:

### User Instruction:
{USER_INSTR}

### Constraints Dictionary:
{DICT_CONSTRAINTS}

Return ONLY the final modified instruction:

Figure 23: Prompt Template for Adding Constraints to Original Instructions

**Prompt Template for Ranking Multiple Instructions** Figure 24 shows the prompt template used for voting-based instruction selection. In this setup, the LLM is presented with four candidate instructions (labeled A–D) for the same task and asked to rank them based on two criteria: clarity of requirements and language fluency. The prompt strictly instructs the LLM to output only the ranked order in a fixed format without any explanation, facilitating consistent and automatable comparison. This template is employed to curate high-quality instruction data by identifying the most effective formulation among alternatives.

---

**Prompt Template for Ranking Multiple Instructions**

###Task:
I will provide four versions of user instructions (labeled A, B, C, D) for the same task. Evaluate them based on:
1. Clarity of requirements: Whether the instruction is unambiguous and directly actionable.
2. Language fluency: Whether the instruction is grammatically correct and easy to understand.

###Output Requirements:
Only output the ranked order in the format: Best > Good > Fair > Worst (e.g., B > A > D > C).
Do not include any explanations or analysis.

###Instructions to Evaluate:
A: {INSTR_A}
B: {INSTR_B}
C: {INSTR_C}
D: {INSTR_D}

###Output:

---

Figure 24: Prompt Template for Ranking Multiple Instructions

**Prompt Template for Ranking Multiple Responses**   To facilitate the selection of the optimal response from a pool of candidates, we designed a prompt for ranking multiple response. This prompt guides multiple models to evaluate and rank the responses based on predefined criteria, ensuring the selection of the optimal response. The structure of the prompt is shown in Figure 25. The prompt template in Figure 25 includes placeholders for the candidate instructions and evaluation criteria. This design ensures that each model provides a structured and comparable assessment of the instructions.

> ### Prompt Template for Ranking Multiple Responses
>
> ###Task: Rank the following 4 responses (A, B, C, D) to the user instruction:
> "{USER_INSTRUCTION}"
>
> ###Evaluation Criteria:
> Instruction Adherence – Does the response follow the constraints of the instruction?
> Helpfulness – Does the response fully address the user's request?
> Accuracy – Is the information correct and reliable?
> Clarity – Is the response well-structured and easy to understand?
> Conciseness – Is the response efficient without unnecessary details?
>
> ###Responses:
> A: {Response_A}
> B: {Response_B}
> C: {Response_C}
> D: {Response_D}
>
> ###Output Format:
> Only provide the ranking in the exact format:
> Best > Good > Fair > Worst (e.g., B > A > D > C)
> Do not include explanations, analysis, or additional text.
> Do not deviate from the ranking format.
>
> ###Output:
> '"

Figure 25: Prompt Template for Ranking Multiple Responses

**Prompt Template for Evaluating Responses**    To systematically assess whether a model response strictly adheres to a specific constraint within an instruction, we utilize a detailed evaluation prompt. This prompt guides the LLM to objectively determine if the constraint is met, focusing solely on the specified constraint rather than the entire instruction. The structure and guidelines of the prompt are designed to ensure precise and unbiased evaluations. The prompt is illustrated in Figure 26.

---
**Prompt Template for Evaluating Response**

Please act as an objective and fair evaluator to analyze the **model response** content and choose "Yes" or "No" to answer whether the subsequent **constraint** is met.
## Please provide an objective and fair answer based on the following judgment rules:
- The **constraint** can be understood as a step-by-step scoring point of the **input instruction**, judging whether a specific part is satisfied. Therefore, you only need to consider whether the requirements stated in the **constraint** are met, without focusing on whether the entire **input instruction** is completely satisfied.
Example:
    - **Input instruction**: "Please generate a poem, and end with 'Generation complete.'"
    - **Model response**: "Moonlight before my bed, perhaps frost on the ground. Lifting my head, I gaze at the bright moon, lowering my head, I think of my hometown."
    - **Constraint**: "The poem is written by Li Bai."
For this constraint, although the model response did not complete the **input instruction** requirement to end with "Generation complete," it met the **constraint** requirement, as the poem is Li Bai's "Quiet Night Thoughts," so the answer should be: "Yes."
For situations where the **constraint** requirements are met, but the entire **input instruction** is not completely satisfied, the answer should be "Yes."
- Yes: Please check whether the **model response** meets the **constraint**, fully understand the meaning of the **constraint**, don't miss small details, focus only on the current **constraint**, don't pay attention to other requirements in the **input instruction**. The response must perfectly and adequately meet the constraint requirements to be evaluated as "Yes." Even if there is a small error or ambiguous content, it cannot be "Yes." There should be no claims like "basically correct," "mostly correct," or "correct under certain conditions." Such cases should all be evaluated as "No."
- No: If the text in the **model response** cannot meet the requirements of the current **constraint** or does not provide information that can be used to verify the constraint. Choose "No."
Example: If the constraint states "The second sentence of the generated text is a compound sentence" but the **model response** has only one sentence. It does not provide relevant information to verify the constraint. Therefore, the answer should be "No."
## Scoring details:
(1) When evaluating whether the model response is itemized, it must have clear bullet points or numbers to be considered itemized and evaluated as "Yes," otherwise evaluate as "No." Using only connecting words such as "first," "then," "second," "finally," etc., cannot be recognized as itemization and should be evaluated as "No."
(2) When evaluating whether the model response uses a certain language (such as Chinese/English) for output, unless the **input instruction** mentions the need to use multiple languages, it must use only that language to be evaluated as "Yes." If multiple languages are mixed (i.e., words from other languages appear), it should be evaluated as "No."
(3) If a **constraint** contains descriptions like "each," "all," etc., the satisfaction of each object needs to be considered. If any object does not meet the requirements, it should be evaluated as "No"; only when all objects meet the requirements can it be evaluated as "Yes."
(4) For **constraints** stating "The model correctly judges" something, when evaluating whether the model correctly judged a certain selection condition, it is necessary to judge whether the model response correctly chose the corresponding branch based on the instruction requirements of different selection branches in the **input instruction**:
## Output format
Return in json format
```json
{{{{
    "Analysis": "xxx",
    "Answer": Yes/No
}}}}
```
## Evaluation information
**Input instruction**
{input}
**Model response**
{output}
**Constraint**
{constraint}
Please analyze and answer whether the **model response** satisfies the **constraint**:

---

Figure 26: Prompt Template for Evaluating Responses

The prompt template in Figure 26 includes the following key components:

- **Objective and Fair Evaluation**: The LLM is instructed to act as an objective and fair validator, analyzing the model response content and choosing "Yes" or "No" to indicate whether the constraint is met.

- **Judgment Rules**: The prompt provides clear rules for evaluation, emphasizing that the constraint should be understood as a specific scoring point of the input instruction. The evaluation should focus solely on whether the requirements stated in the constraint are met, without considering the overall satisfaction of the input instruction.

- **Examples**: The prompt includes examples to illustrate how to apply the judgment rules. For instance, even if the model response does not fully satisfy the input instruction, it can still meet the constraint, resulting in a "Yes" answer.

- **Scoring Details**: The prompt outlines specific scoring details for various types of constraints, such as itemization, language usage, and the presence of specific elements in the response.

- **Output Format**: The LLM is instructed to return the evaluation in a structured JSON format, including an analysis and a clear "Yes" or "No" answer.

This prompt ensures that each constraint is evaluated rigorously and consistently, providing a reliable method for assessing the adherence of model responses to specific constraints.

## LLM USAGE STATEMENT

LLMs did not contribute to the research ideation, experimental design, analysis, or manuscript writing. All conceptual and textual contributions are solely attributable to the authors.

