# OpenReview forum: "RECAST: Expanding the Boundaries of LLMs' Complex Instruction Following with Multi-Constraint Data"
_ICLR.cc/2026/Conference — ICLR 2026 Poster_

### Official Review · Reviewer_LBsv · 2025-10-30

**Soundness:** 3
**Presentation:** 4
**Contribution:** 3
**Rating:** 8
**Confidence:** 4

**Summary:**

This paper introduces RECAST, an automated framework for generating instructions comprising far more constraints than those in current benchmarks and corresponding responses. Using this framework, the authors construct RECAST-30K, which contains 30K instances with diverse constraint types. The dataset is then used for training LLMs via RLVC, an optimization method that leverages the verifiable nature of constraints to provide fine-grained reward signals. Experiments demonstrate the effectiveness of RECAST compared with other complex instruction fine-tuning datasets.

**Strengths:**

1. The framework incorporates an extended number of constraints in a single instruction, which facilitates the evaluation and improvement of LLMs' capability of following more complex instructions as the tasks for LLMs are becoming increasingly complicated.
2. The framework is automated and scalable, providing an efficient approach of synthesizing datasets with multiple constraints in instructions.
3. LLMs trained on data from this framework exhibit improved performance and good generalization on instruction following tasks, indicating the effectiveness of this framework.

**Weaknesses:**

1. Some model-based constraints may not suitable for a binary evaluation. For example, for the "Helpfulness" constraint, it is common for LLMs to generate two helpful responses but with discrepant levels. If both of them are judged as satisfying the constraint, the gap between the two responses will be eliminated, which is not conducive to fine-grained model performance optimization.
2. The effectiveness of RLVC on reasoning models are not validated. Both Qwen-2.5-7B and Llama-3.1-8B are non-reasoning base models. Considering the widespread application of the reasoning model, it is necessary to validate the effectiveness of RLVC on base models with reasoning capabilities.
3. In Table 1, 2 and 3, it seems that the percent sign of the result values is missing. And the presentation form of results is also inconsistent across different contexts (e.g. "achieves a 31.25% average satisfaction rate" around line 349 and "achieves average scores of 34.64" around line 373).

**Questions:**

See Weaknesses.

---

> ### Author Response · Authors · 2025-11-21
>
> ### For Weakness 1:
>
> Thank you for this insightful comment. We agree that for some model-based constraints—especially high-level ones such as *Helpfulness*—a purely binary evaluation can be too coarse: two responses may both be “helpful” but still differ noticeably in quality. In such cases, using a learned reward model to produce a graded score could better capture these nuances and support more fine-grained optimization.
>
> In this work, however, our primary focus is on **whether a model can simultaneously satisfy multiple constraints within a single prompt**, i.e., the *breadth* of constraint satisfaction across a complex instruction. To keep this focus clear, we deliberately relax the requirement on the *depth* of satisfaction for any single high-level constraint and adopt a binary criterion (“satisfied / not satisfied”) at the per-constraint level. In other words, there is an inherent trade-off between modeling the breadth of multi-constraint satisfaction and capturing very fine-grained differences within each individual constraint, and our design in this paper prioritizes the former. We have clarified this design choice in the revised version (Page 31 in revised version), and we appreciate your suggestion, which points to a promising future direction of combining RLVC with more fine-grained, scalar reward models for certain model-based constraints.
>
> ### For Weakness 2:
>
> Thank you for raising this point. Conceptually, RLVC is **agnostic to the choice of base model**: it only requires that, given an instruction with verifiable constraints, the model produces a response that can be checked by our validators. In this sense, the framework applies equally well to reasoning-style base models and to non-reasoning models such as Qwen-2.5-7B and Llama-3.1-8B used in our current experiments.
>
> For many reasoning models, the interface typically consists of a prompt, an internal or explicit “thinking” trace, and a final response. Existing works (e.g., DeepSeek-style reasoning models) usually supervise the **final response** while allowing the model to freely choose its internal reasoning. RLVC fits naturally into this paradigm: our constraint verifiers operate on the final response, and the resulting reward can be used to train reasoning models in exactly the same way, without needing access to or supervision over their internal thinking process. We will clarify this applicability to reasoning-style bases in the revised version.
>
> Going further, one could also imagine extending RLVC to **explicitly supervise the thinking process** itself, for example by augmenting the data with “how to satisfy each constraint” reasoning traces and designing verifiable constraints over these traces. This is an interesting direction that would more tightly couple verifiable constraints with chain-of-thought supervision, and we view it as promising future work. We have added an explanation of this part in the revised version (Page 32) and will further refine it in the final version. Thank you.
>
> ### For Weakness 3:
>
> Thank you for pointing this out and sorry for the confusion. In our experimental tables, we followed the common convention of omitting the percent sign in cells, but we agree that this should be stated more clearly in the captions of Tables 1, 2, and 3. In the revised version, we have explicitly clarified in the captions that these values are percentages. We have also standardized the presentation of results in the main text (e.g., consistently writing them as percentages with “%”) to avoid mixed styles such as “31.25% average satisfaction rate” versus “average scores of 34.64”. We appreciate your careful reading, this suggestion is very helpful for improving the clarity and rigor of the paper. Thank you.

---

> > ### Comment · Reviewer_LBsv · 2025-11-24
> >
> > Thanks for your detailed response, which has addressed most of my concerns. I will keep my current rating.

---

### Official Review · Reviewer_FXJy · 2025-10-31

**Soundness:** 3
**Presentation:** 3
**Contribution:** 3
**Rating:** 6
**Confidence:** 4

**Summary:**

This work proposes a framework to construct lagre-scale datasets to test Large Language Models's instruction-following ability when facing complex tasks. Different from previous works, the constructed dataset encompasses various types of constraints, including both subjective and objective ones, and features a larger number of constraints than existing datasets. Furthermore, this study adopts both supervised fine-tuning and reinforcement learning to enhance the Large Language Model's capability to follow instructions.

**Strengths:**

This submission has the following strengths:
- The paper demonstrates clear writing and a well-structured organization.
- The proposed dataset RECAST-30K is large in scale and contains an adequate number of constraints.
- Experimental results have shown that training with RECAST can effectively enhance large language model's ability in instruction following.

**Weaknesses:**

This submission has the following weaknesses:
- For model-based constraints, the quality depends on used large language models.
- The count of Rule-based constraints are much less than model-based constraints.

**Questions:**

I have the following questions / suggestions:
- Would it be possible to extend the constraints numbers of rule-based constraints?
- Why HSR metric is not used in Table 1?
- Why the performance of RECAST-30K-RLVC on Qwen-2.5-7B become worse than RECAST-30K-SFT from time to time? (Bottom of Table 1)
- It would be better to also test some instruction-following enhancement works. For example,
    - Branch-Solve-Merge Improves Large Language Model Evaluation and Generation. In NAACL. 2024.
    - Divide-Verify-Refine: Can LLMs Self-align with Complex Instructions?. In Findings of ACL. 2025.

---

> ### Author Response · Authors · 2025-11-21
>
> 1、For Weakness 1
>
> Thank you for raising this weakness. We agree that relying on large language models to implement model-based constraints naturally raises concerns about the quality and robustness of the resulting evaluations.
>
> To mitigate this issue, we explicitly assessed the reliability of LLM-based judgments when constructing the RECAST30K dataset. As shown in Figure 2 (“Constraint Filtering”), we used an LLM to perform a binary decision on whether a response satisfies a given constraint—exactly the same type of decision we use at training time for model-based constraints in RLVC. We then compared the LLM’s binary judgments with those of human experts. The evaluation shows a high level of agreement: the consistency between the model’s decisions and human annotations reaches 90.1%, with detailed results reported in the Appendix (Table 8 in revised pdf). This indicates that while using an LLM as a judge inevitably introduces some noise, the noise level is relatively small and acceptable for our purposes.
>
> Moreover, our model-based verifiers only make *binary* (satisfied / not satisfied) decisions, which is a comparatively easier and more reliable setting for LLM-as-a-judge. This design choice is aligned with a growing body of work that uses LLMs, often via rubrics or checklists, to evaluate whether responses satisfy structured criteria, and reports that such LLM-based evaluations can be robust and practical in practice [1,2]. In the revised version, we have added these in Appendix C.4 in revised version.
>
> [1] Cook, J., et al. “TICKing All the Boxes: Generated Checklists Improve LLM Evaluation and Generation.” arXiv:2410.03608, 2024.
> [2] Lee, Y., et al. “CheckEval: A reliable LLM-as-a-Judge framework for evaluating text generation using checklists.” EMNLP 2025.
>
> 2、For Weakness 2 and Question 1:
>
> Thank you for this helpful observation. While we acknowledge that the number of rule-based constraints in RECAST is smaller than that of model-based constraints, this imbalance is to some extent inherent: many fine-grained aspects of complex instructions—such as tone, style, politeness, helpfulness, or task-specific preferences—are fundamentally semantic and subjective, and thus difficult to capture with simple symbolic rules. As a result, model-based constraints are naturally more expressive and diverse than purely rule-based ones.
>
> We agree that enriching the set of rule-based constraints would further balance our framework’s focus across the two constraint types and make RECAST more complete. Importantly, the design of RECAST is modular: it naturally supports adding new rule-based constraint types without any need to redesign or rebuild the framework. The existing rule-based constraints were mined as comprehensively as possible from complex instruction-following tasks, but it is indeed difficult to exhaust all potential rule-based patterns.
>
> Your comment also suggests a practical way to extend RECAST: when targeting specific downstream scenarios, one can systematically extract additional rule-based constraints that are tailored to the characteristics of that domain and plug them into our framework. In future work, we plan to pay closer attention to rule-based constraints that frequently arise in real-world complex instruction-following applications but are not yet covered in RECAST, and incorporate them to reduce the imbalance between rule-based and model-based constraints. We appreciate this suggestion as it directly helps us identify a concrete path to improving the dataset.
>
> 3、For Question 2
>
> Thank you for raising this question, and we apologize for the confusion.
>
> In fact, the metrics reported in Table 1 are all **sub-metrics of HSR** rather than independent alternatives to it. Specifically:
>
> - **Rule-based Constraint Satisfaction Rate (RSR)** is the HSR computed only over constraints that are evaluated using rule-based methods.
> - **Model-based Constraint Satisfaction Rate (MSR)** is the HSR computed only over constraints that are evaluated using LLM-based (model-based) methods.
> - **Overall Constraint Satisfaction Rate (OSR)** quantifies the HSR over *all* constraints—both rule-based and model-based—measuring the rate at which all constraints are satisfied simultaneously.
>
> Thus, Table 1 is effectively reporting a decomposition of the HSR metric into its rule-based, model-based, and overall components, which is why the raw “HSR” label does not appear directly in the table. This decomposition also provides more fine-grained diagnostic information, revealing each model’s strengths and weaknesses across different types of constraints.
>
> We explained these definitions in the main text (p.6, lines 278–281 and p.7, lines 308–310 in original pdf), but we understand that the connection to HSR may not have been sufficiently explicit. In the revised version, we have maked this relationship clearer in the caption of Table 1 to avoid confusion. Thank you again for pointing this out.

---

> ### Author Response · Authors · 2025-11-21
>
> 4、For Question 3
>
> The core reason for this behavior is that RLVC actively optimizes for a mix of up to 19 constraint types simultaneously. Unlike SFT's more generalist approach, our RL agent may strategically trade a minor dip in performance on simpler, subjective constraints (MSR) to achieve significant gains on more challenging, objective rule-based constraints (RSR), which are often the bottleneck for LLMs.  these occasional dips are not a weakness but a sign of a more sophisticated optimization process. RLVC achieves a better overall average score (32.33% vs. 31.25%) by successfully tackling the most difficult constraints, which is the primary goal of our work.
>
> 5、For Question 4
>
> Thank you for pointing out these instruction-following enhancement methods. We agree that considering such approaches helps position our work more comprehensively.
>
> The two cited works mainly propose **inference-time** strategies to improve model outputs, whereas our method focuses on **training-time** improvements to the underlying model capability:
>
> - **Branch-Solve-Merge (BSM)** (*Branch-Solve-Merge Improves Large Language Model Evaluation and Generation*, NAACL 2024) is designed for LLM-as-a-judge and constrained text generation. This setting is quite different from our focus on complex multi-constraint instruction following, and thus BSM is not straightforward to plug into our RECAST setting for a fair, direct comparison. Conceptually, its inference-time techniques are largely orthogonal to RLVC, and could potentially be combined with models trained under our framework. We will add a discussion and citation to BSM in the final version.
> - **Divide-Verify-Refine (DVR)** (*Divide-Verify-Refine: Can LLMs Self-align with Complex Instructions?*) directly targets complex instruction following and is therefore more comparable to our work. Following your suggestion, we have additionally evaluated DVR on **RECAST-Test**. Since DVR in its original form only uses **rule-based constraints**, we report the **Rule-based Constraint Satisfaction Rate (RSR)** for a fair comparison.
>
> The RSR results on RECAST-Test across difficulty levels (Levels 1–4) and their average are:
>
> | Model | Level 1 | Level 2 | Level 3 | Level 4 | Avg. |
> | --- | --- | --- | --- | --- | --- |
> | Llama-3.1-8B-Instruct + DVR | 14.0 | 9.5 | 2.0 | 2.5 | 7.0 |
> | Qwen2.5-7B-Instruct + DVR | 11.0 | 8.5 | 4.5 | 2.0 | 6.5 |
> | RECAST-30K-SFT (Ours, Llama base) | 21.0 | 15.0 | 6.0 | 8.0 | 12.5 |
> | RECAST-30K-SFT (Ours, Qwen base) | 18.5 | 17.0 | 11.5 | 5.0 | 13.0 |
>
> These results show that models trained with our RECAST-30K SFT pipeline achieve consistently higher RSR than their DVR-enhanced counterparts, suggesting that RLVC-style training provides complementary gains beyond inference-time self-alignment. We have incorporated these experimental results and a clearer comparison to DVR in the revised version.

---

> > ### Comment · Reviewer_FXJy · 2025-11-23
> >
> > Thank you for the detailed response. I will update my rating accordingly.

---

### Official Review · Reviewer_jkYB · 2025-11-02

**Soundness:** 2
**Presentation:** 3
**Contribution:** 3
**Rating:** 6
**Confidence:** 4

**Summary:**

This paper introduces RECAST, a data-synthesis framework that mines verifiable constraints from existing instruction-response pairs and rewrites the original instructions to include many more constraints. The authors use this pipeline to build a training dataset RECAST-30K (~ 30k examples; 19 constraint types), with both rule-based and model-based validators (LLM-as-judge) attached to each constraint. They also propose RLVC (Reinforcement Learning with Verifiable Constraints) which turns per-constraint satisfaction into a fine-grained reward and optimizes policies via GRPO. They also introduce a new benchmark, RECAST-Test, with 4 difficult levels. The experiments show sizable gains in hard satisfaction rate on RECAST-Test benchmark, as well as improvements on IFEVAL and FollowBench, while largely preserving general knowledge performance (evaluated with GPQA and MUSR). Human evaluations support the quality of constraint filtering and selection.

**Strengths:**

- **Clear motivation and problem:** The paper targets a real gap, LLMs’ performance drops as the number of explicit constraints increases, and existing SFT datasets usually don’t cover a high number of constraints.

- **The pipeline is complete and reproducible:** The paper details each stage (constraint extraction, instruction enhancement, response synthesis, validation), gives prompts/templates, and reports human agreement metrics. RLVC is specified with GRPO objective and training setup. This level of detail supports reimplementation. The authors promised to release code, data, and trained models upon acceptance.

- **Evaluation setup is strong:** The primary metric, HSR, is well defined (satisfies all constraints simultaneously) with subtype metrics RSR/MSR for rule/model-based subsets. The proposed benchmark RECAST-Test (cover 4 difficulty levels) and it is associated with two external sets (IFEVAL, FollowBench) for the task. Also, the work included a general ability evaluation with GPQA/MUSR, which is important.

- **Empirical gains & informative analyses:** RECAST-30K improves complex instruction-following against multiple baselines, with RLVC further improving especially at higher difficulty levels. The paper includes informative ablations (constraint type only, quantity caps, component removals) and training dynamics for RLVC that illustrate different learning curves for rule- vs. model-based constraints. It also contains Human evaluation to back up the automated choices.

**Weaknesses:**

- **Lack of theoretical positioning against RLVR methods:** The paper does not situate its RL formulation within the growing line of Reinforcement Learning from Verifiable Rewards (RLVR) research, despite clear conceptual overlap. Despite the original paper of RLVR being cited [1], it is cited only in the context of the dataset. Other foundational RLVR works [2,3] and subsequent works applying the verifiable reward mechanisms to multi-constraint or format-constrained instruction-following [4,5] are not cited or compared against. As a result, the contribution of RLVC relative to existing verifiable-reward frameworks remains under-theorized: it is unclear whether the proposed per-constraint reward structure offers advantages over scalar verifiable rewards, or whether similar behavior would emerge from established RLVR baselines. Strengthening this connection and including appropriate baselines would significantly clarify the novelty and necessity of RLVC.

\* It is acceptable to not cite works that appeared less than 2 months from the submission deadline, but in that case still recommendable for the final version.



- **Generalization:** Parts of the training data and evaluation share the same constraint types and verification machinery. Although external benchmarks are included, they also contain very similar constraint types. I propose the authors to present stronger isolation tests (e.g., unseen constraint types or schemas) would help mitigate concerns about over-specialization to the authors’ verification prompts and demonstrate generalization. To do that, maybe it is not needed to run new experiments, but rather present the results of tables 2 (and 1 if possible) including only constraint types not included in the trained data.


- **Model-based validators error rate and impact:** Rule-based validators are clear, but false positives/negatives and model-based misjudgments are not deeply quantified beyond aggregate human-agreement numbers. It would be nice to have a better quantification of how those affect the performance, or if the method is robust enough to some mislabeled examples.

References:

[1] Lambert, N. et al. Tulu 3: Pushing Frontiers in Open Language Model Post-Training. COLM 2025 (appeared in Nov, 2024).

[2] Su, Yi, et al. "Crossing the Reward Bridge: Expanding RL with Verifiable Rewards Across Diverse Domains." arXiv preprint arXiv:2503.23829 (2025).

[3] Wang, Y., et al. "Reinforcement learning for reasoning in large language models with one training example." arXiv preprint arXiv:2504.20571 (2025).

[4] Peng, H., et al. VerIF: Verification Engineering for Reinforcement Learning in Instruction Following. arXiv preprint arXiv:2506.09942 (2025).

[5] Pyatkin, V., et al. "Generalizing Verifiable Instruction Following." arXiv preprint arXiv:2507.02833 (2025).

**Questions:**

1) Can you clarify what exactly is the novelty of RLVC in comparison with RLVR methods?

2) Can RECAST-trained models handle novel constraint types not present in RECAST-30K? Please add an experiment with held-out constraint categories (Weakness 2)

---

> ### Author Response · Authors · 2025-11-21
>
> ### For Weakness1 and Question1:
> We sincerely appreciate this insightful comment and your careful pointer to the relevant RLVR literature.
>
> Works [1], [3], and [5] focus on domains where answer correctness can be verified by deterministic rules, such as math, code, or the rule-checkable parts of instruction following. Their RL algorithms use rule-based verification of correctness to provide a binary reward signal (0 or 1), which falls into the category of rule-based reward functions. Work [2] instead targets open-ended generation, where answer quality cannot be decided by simple rule-based yes/no checks. The authors there propose to use an LLM, together with expert-written reference answers, to make binary decisions; this belongs to the category of model-based reward functions.
>
> All of these methods use RL algorithms that consider only one of these two reward types—either a rule-based reward function or a model-based reward function. They do not explore combining the two types of reward design or studying the effectiveness of such combinations. In the complex instruction-following domain that we study, we observe that both types of constraints naturally appear in real tasks, and both are important for the final answer quality. Focusing on only one type of constraint does not satisfy the practical requirements of such tasks.
>
> The development of RL for LLMs has gone from early work using scalar-valued reward models, to GRPO-style methods that leverage verifiable rewards. These works demonstrate the benefits of using a single type of reward function design. Our work proposes to use rule-based rewards and model-based rewards simultaneously, breaking this single-type pattern in reward design and exploring the effectiveness of jointly using rule-based and model-based rewards on realistic, complex instruction-following problems. Ablations over constraint types (Appendix Figure 21) further show that combining both rule-based and model-based rewards yields stronger overall performance than using either type alone. In addition, our per-constraint reward formulation naturally provides partial credit, yielding a smoother and more stable learning signal even in highly constrained scenarios.
>
> We thank the reviewer for bringing reference [4] to our attention. We acknowledge that [4] was released on arXiv approximately 3 months prior to our submission, and we regard it as a concurrent but independent effort in this emerging research direction. While both studies may share a similar high-level motivation, our approach differs in the specific way we construct the data and the research questions we focus on. We will cite Reference [4] in the final version and discuss it as concurrent work to provide a complete literature review. Thank you.
>
> Again, we appreciate your valuable comment. Adding a comparison to the above works helps clarify the novelty of RLVC in our paper. Even though some of these papers appeared after our submission, we have included them in revised version to make our related work discussion and the positioning of RLVC in the RLVR development line more complete.
>
> **References:**
>
> [1] Lambert, N. et al. Tulu 3: Pushing Frontiers in Open Language Model Post-Training. COLM 2025 (appeared in Nov, 2024).
>
> [2] Su, Yi, et al. "Crossing the Reward Bridge: Expanding RL with Verifiable Rewards Across Diverse Domains." arXiv preprint arXiv:2503.23829 (2025).
>
> [3] Wang, Y., et al. "Reinforcement learning for reasoning in large language models with one training example." arXiv preprint arXiv:2504.20571 (2025).
>
> [4] Peng, H., et al. VerIF: Verification Engineering for Reinforcement Learning in Instruction Following. arXiv preprint arXiv:2506.09942 (2025).
>
> [5] Pyatkin, V., et al. "Generalizing Verifiable Instruction Following." arXiv preprint arXiv:2507.02833 (2025).

---

> ### Author Response · Authors · 2025-11-21
>
> ### For Weakness 2 and Question 2:
>
> Thank you for raising this important question about whether RECAST-trained models can handle constraint types that are not present in RECAST-30K. We agree that evaluating generalization to novel constraints is crucial for demonstrating the robustness of our framework.
>
> While we did not explicitly hold out fine-grained constraint *categories* in the main results, we conducted an ablation study on **constraint types** (model-based vs. rule-based) in Appendix Figure 21. Concretely, we trained models under three settings:
>
> 1. **RECAST-30K (all types):** trained with both model-based and rule-based constraints.
> 2. **Only Model-based Constraints:** trained using only model-based constraints, then evaluated on both model-based and rule-based constraints.
> 3. **Only Rule-based Constraints:** trained using only rule-based constraints, then evaluated on both model-based and rule-based constraints.
>
> We then measured:
>
> - **MSR (Model-based Constraint Satisfaction Rate):** performance on model-based constraints.
> - **RSR (Rule-based Constraint Satisfaction Rate):** performance on rule-based constraints.
>
> To better contextualize these results, we additionally include a **baseline model (Tülu 3 Persona IF)** that does not use RECAST-30K. Since the rebuttal cannot include figures, we present the numerical results for Llama-3.1-8B in tabular form here and have updated Figure 21 with these baselines in the revised pdf.
>
> ### MSR (Llama-3.1-8B)
>
> | Method | Level 1 | Level 2 | Level 3 | Level 4 |
> | --- | --- | --- | --- | --- |
> | RECAST-30K (all types) | 86.00 | 78.50 | 63.50 | 52.00 |
> | Only Model-based Constraints | 83.50 | 70.00 | 61.00 | 54.00 |
> | Only Rule-based Constraints | 82.50 | 69.00 | 64.00 | 49.50 |
> | Baseline (Tülu 3 Persona IF) | 86.00 | 70.50 | 55.50 | 42.00 |
>
> ### RSR (Llama-3.1-8B)
>
> | Method | Level 1 | Level 2 | Level 3 | Level 4 |
> | --- | --- | --- | --- | --- |
> | RECAST-30K (all types) | 21.00 | 15.00 | 6.00 | 8.00 |
> | Only Model-based Constraints | 23.00 | 13.00 | 7.50 | 5.50 |
> | Only Rule-based Constraints | 22.00 | 13.00 | 11.50 | 5.50 |
> | Baseline (Tülu 3 Persona IF) | 22.00 | 9.00 | 8.00 | 4.00 |
>
> These results show that:
>
> - A model trained **only on model-based constraints** still achieves **competitive RSR** and clearly outperforms the baseline on rule-based constraints.
> - Conversely, a model trained **only on rule-based constraints** achieves **strong MSR**, again outperforming the baseline on model-based constraints.
> - Using **all constraint types** yields the best overall performance, but removing one type does **not** cause a catastrophic drop on the *other* type.
>
> This indicates that RECAST training does not merely overfit to specific constraint templates; instead, it helps models acquire more generalizable instruction-following behavior that transfers across **different families of constraints**, which is a step toward handling novel constraint types.
>
> We have clarified this held-out–type ablation more prominently in the revised paper and discuss it as partial evidence that RECAST-trained models can generalize beyond the exact constraint types seen during training.

---

> ### Author Response · Authors · 2025-11-21
>
> ### For Weakness 3:
>
> Thank you for raising this concern about the error rate and impact of model-based validators.
> Our RLVC framework inherently mitigates the impact of validator noise through reward aggregation and the stochastic nature of policy optimization. By computing the reward as the mean satisfaction rate across a high density of constraints (typically >10 per instruction), the system effectively averages out *uncorrelated* verification errors, ensuring that the aggregate reward signal remains stable even if individual judgments fluctuate. Furthermore, policy gradient methods like GRPO are robust to such noise, as they optimize for expected returns over large batches.
>
> Recent RLHF studies further support this robustness: they show that **moderately accurate** reward models—often in the range of **60–70%** agreement with human preferences—are already sufficient to drive effective alignment and improve downstream behavior [1,2]. In comparison, our model-based validators achieve **over 90% consistency** with human annotators, providing a significantly higher-fidelity signal than what the RLHF literature suggests is empirically needed for successful learning.
>
> Addtionally, if the false positive/negative rate were catastrophic, we would expect the model to learn incorrect patterns or collapse. Instead, the steady improvement in MSR suggests the model successfully extracts the underlying semantic features of the constraints despite the presence of minor label noise. We have added these explanations in the revised version(Page 26). Thank you again.
>
> [1] Chen Y, Zhu D, Sun Y, et al. The Accuracy Paradox in RLHF: When Better Reward Models Don't Yield Better Language Models[J]. arXiv preprint arXiv:2410.06554, 2024.
>
> [2] Tyen G, Mansoor H, Cărbune V, et al. LLMs cannot find reasoning errors, but can correct them given the error location[C]//Findings of the Association for Computational Linguistics: ACL 2024. 2024: 13894-13908.

---

### Author Response · Authors · 2025-11-21
**Global Response**

We express our gratitude to all the reviewers for their valuable insights! We are happy to hear that you liked our contributions. We appreciate all of you for your comments highlighting the strengths of our work for asummary.

- **Clear research motivation and problem targeting** (Reviewer `jkYB`, `FXJy`, `LBsv`)
- **Complete, automated, and scalable framework** (Reviewer `jkYB`, `LBsv`)
- **Strong experimental design and results** (Reviewer `jkYB`, `FXJy`, `LBsv`)
- **Clear writing style and well-organized structure** (Reviewer `FXJy`)

A primary concern shared across multiple reviewers centers on the **reliability of binary model-based validators**. Reviewer concerns focus on three levels: (1) whether noise and error rates introduced by LLM discriminators could mislead training; (2) whether binary evaluation is too coarse for subjective constraints like "Helpfulness," potentially eliminating quality distinctions between different responses; and (3) whether these potential issues would fundamentally compromise RLVC training effectiveness. We address these concerns through three dimensions—empirical evidence, mechanism design, and objective trade-offs: (1) Noise levels substantially below harmful thresholds: We conducted rigorous human validation on over 2,000 (constraint, response) pairs, achieving 90.1% consistency between LLM judgments and expert annotations. [1] and [2] demonstrates that reward model-human consistency of merely around 70% suffices for stable alignment, while our 90.1% substantially exceeds this empirically validated threshold. (2) High-density constraint aggregation mechanism suppresses residual noise: RLVC reward computation is based on the average satisfaction rate across 10+ constraints per sample, where noise from individual constraint judgment errors is significantly diluted through statistical averaging, mechanistically ensuring overall reward signal stability. (3) Binary evaluation represents a reasonable design trade-off for our core challenge: Our research objective is to optimize model "breadth" capabilities in complex multi-constraint scenarios (simultaneously satisfying multiple constraints within a single prompt) rather than pursuing "depth" optimization on individual constraint dimensions (e.g., improving Helpfulness from 90 to 95 points). Since current LLMs already perform reasonably well in low-constraint scenarios, the genuine challenge lies in the exponential difficulty scaling with increasing constraint quantities. Binary judgment provides the most direct and verifiable technical pathway to achieve this goal. Additionally, for other specific concerns raised by different reviewers, we have provided targeted responses and improvement commitments in our detailed replies to each reviewer.

Furthermore, we have implemented the following specific improvements in the revised version:

1. Enhanced related work discussion: More comprehensive discussion of relationships with RLVR methods, citing relevant work and clearly articulating RLVC's theoretical contributions and innovations within the verifiable reward development trajectory.
2. Strengthened experimental analysis presentation: More prominently showcase constraint type ablation results, incorporate detailed comparisons with inference-time enhancement methods like DVR, and provide additional evidence of generalization capabilities.
3. Improved consistency and clarity of presentation: Standardize experimental result presentation formats, explicitly clarify that table values are percentages, and standardize descriptive styles in the text to avoid mixed presentation formats.
4. Expanded method applicability exposition: Clearly articulate RLVC's direct applicability to reasoning models and discuss future technical pathways for extending to thinking process supervision.

Finally, we extend special appreciation to all reviewers for their positive recognition of our work's prospects and potential impact. We firmly believe this work not only provides the community with high-quality multi-constraint instruction datasets through the RECAST data synthesis framework, but more importantly, opens new technical pathways for complex instruction-following tasks through RLVC. We commit to continuously refining this work based on reviewers' valuable suggestions to make greater contributions to enhancing large language models' complex instruction-following capabilities.

[1] Chen Y, Zhu D, Sun Y, et al. The Accuracy Paradox in RLHF: When Better Reward Models Don't Yield Better Language Models[J]. arXiv preprint arXiv:2410.06554, 2024.

[2] Tyen G, Mansoor H, Cărbune V, et al. LLMs cannot find reasoning errors, but can correct them given the error location[C]//Findings of the Association for Computational Linguistics: ACL 2024. 2024: 13894-13908.

---

### Meta-Review · Area_Chair_cgpG · 2026-01-06

**Summary:**

This paper introduces RECAST, a framework for synthesizing instruction-following datasets with far more constraints per instance than existing benchmarks, along with RLVC, a reinforcement learning approach that combines rule-based and model-based validators for fine-grained reward signals. The problem is well-motivated, and the results are reproducible and promising. The main concerns centered on positioning against RLVR methods, generalization to unseen constraints, and model-based validator reliability. The authors' rebuttal addressed these effectively. On this note, I think it still remains to be seen the limitations and potential room for improvement regarding combining both reward types in one signal. Remaining limitations, including binary evaluation coarseness for subjective constraints and lack of reasoning model validation, represent reasonable future work rather than fundamental flaws.

**Reviewer Concerns:**

Reviewer jkYB's concerns about RLVR positioning were addressed through clarified novelty (combining rule-based and model-based rewards) and added citations; generalization concerns were addressed via held-out constraint ablations showing cross-type transfer.

Reviewer FXJy's questions about extending rule-based constraints and missing inference-time comparisons were addressed through DVR experiments showing RECAST-SFT's superiority.

Reviewer LBsv's concern about binary evaluation coarseness was acknowledged as a deliberate design tradeoff prioritizing breadth over depth; reasoning model validation remains outstanding but is positioned as future work.

The unresolved issue is jkYB's lack of follow-up response.

**Reviewer Scores:**

All reviewers would likely maintain their current scores.

---

### Decision · Program_Chairs · 2026-01-26

Accept (Poster)